# Schwann cells, but not Oligodendrocytes, Depend Strictly on Dynamin 2 Function

Daniel Gerber[1†], Monica Ghidinelli[1†], Elisa Tinelli[1], Christian Somandin[1], Joanne Gerber[1], Jorge A Pereira[1], Andrea Ommer[1], Gianluca Figlia[1], Michaela Miehe[1], Lukas G Nägeli[1], Vanessa Suter[1], Valentina Tadini[1], Páris NM Sidiropoulos[1], Carsten Wessig[2], Klaus V Toyka[2], Ueli Suter[1]*

[1]Department of Biology, Institute of Molecular Health Sciences, Swiss Federal Institute of Technology, ETH Zurich, Zurich, Switzerland; [2]Department of Neurology, University Hospital of Würzburg, University of Würzburg, Würzburg, Germany

**Abstract** Myelination requires extensive plasma membrane rearrangements, implying that molecules controlling membrane dynamics play prominent roles. The large GTPase dynamin 2 (DNM2) is a well-known regulator of membrane remodeling, membrane fission, and vesicular trafficking. Here, we genetically ablated *Dnm2* in Schwann cells (SCs) and in oligodendrocytes of mice. *Dnm2* deletion in developing SCs resulted in severely impaired axonal sorting and myelination onset. Induced *Dnm2* deletion in adult SCs caused a rapidly-developing peripheral neuropathy with abundant demyelination. In both experimental settings, mutant SCs underwent prominent cell death, at least partially due to cytokinesis failure. Strikingly, when *Dnm2* was deleted in adult SCs, non-recombined SCs still expressing DNM2 were able to remyelinate fast and efficiently, accompanied by neuropathy remission. These findings reveal a remarkable self-healing capability of peripheral nerves that are affected by SC loss. In the central nervous system, however, we found no major defects upon *Dnm2* deletion in oligodendrocytes.
DOI: https://doi.org/10.7554/eLife.42404.001

*For correspondence:
ueli.suter@biol.ethz.ch

†These authors contributed equally to this work

**Competing interests:** The authors declare that no competing interests exist.

## Introduction

Motor, sensory, and cognitive functions of the nervous system require rapid and refined impulse propagation. In jawed vertebrates, this is achieved by covering axons with myelin (*Weil et al., 2018*). Myelin is formed by consecutive wrappings of the plasma membrane of Schwann cells (SCs) in the peripheral nervous system (PNS) and of oligodendrocytes (OLs) in the central nervous system (CNS). Disturbing myelin can lead to disorders in the PNS and/or CNS, including peripheral neuropathies, multiple sclerosis, and leukodystrophies (*Nave and Werner, 2014*; *Saporta and Shy, 2013*). Among the inherited peripheral neuropathies, Charcot-Marie-Tooth disease (CMT) is characterized by distal muscle weakness and atrophy predominantly in the lower extremities, distal sensory loss, and skeletal deformations such as *pes cavus* (*Saporta and Shy, 2013*). Genetically, CMT is heterogeneous (*Berciano, 2011*; *Gutmann and Shy, 2015*; *Suter and Scherer, 2003*). Among the CMTs, a dominant intermediate form (DI-CMTB) is caused by mutations at the dynamin 2 (*DNM2*) locus (*Claeys et al., 2009*). In addition, mutations in *DNM2* have been associated with other human pathologies, including thrombocytopenia and hematopoietic disorders, for example neutropenia and early T-cell precursor acute lymphoblastic leukemia (*Claeys et al., 2009*; *Zhang et al., 2012*; *Züchner et al., 2005*), and centronuclear myopathy (*Catteruccia et al., 2013*; *Jungbluth and Gautel, 2014*; *Romero, 2010*). DNM2 is a large GTPase that belongs to the dynamin superfamily (*Ferguson and De Camilli, 2012*). The classical dynamin family comprises DNM1, DNM2, and DNM3, differentially expressed throughout the body. DNM1 is found mainly in neurons, DNM2 is

ubiquitously present, and DNM3 is mainly expressed in brain, testis, and lung (*Ferguson et al., 2007*; *Raimondi et al., 2011*). The three isoforms play overlapping roles in membrane fission and endocytosis, in remodeling of mitochondria, lysosomes, endoplasmic reticulum, and Golgi apparatus, as well as in actin and tubulin dynamics (*Durieux et al., 2010*; *Ferguson and De Camilli, 2012*; *Raimondi et al., 2011*), although different properties between isoforms have been described during membrane fission (*Liu et al., 2011*). Complete loss of DNM2 in mice results in early embryonic lethality (*Ferguson et al., 2009*), highlighting the crucial importance of DNM2 in development. Some subsequent work addressed specific DNM2 functions and requirements at the cell type-specific level. These studies include specific ablation of *Dnm2* in mouse skeletal muscle, which resulted in muscle fiber atrophy and loss, lipid droplet accumulation, mitochondrial aberrations, defective neuromuscular junctions, and peripheral nerve degeneration (*Tinelli et al., 2013*). In lymphocytes, loss of *Dnm2* led to reduced proliferation, triggered by reduced T-cell receptor endocytosis and decreased mTORC1 signaling (*Willinger et al., 2015*). In the megakaryocyte lineage, *Dnm2* deletion delayed megakaryocyte maturation in the bone marrow, resulting in macrothrombocytopenia and a reduction in circulating platelet cells (*Bender et al., 2015*). In germline cells, *Dnm2* ablation caused male sterility due to cell cycle arrest, increased cell death, and reduced proliferation (*Redgrove et al., 2016*). However, not all cells are equally dependent on DNM2. For instance, specific ablation of *Dnm2* did not affect neuronal development in the auditory brainstem, but it exacerbated defects of *Dnm1-Dnm3* double knockout mice (*Fan et al., 2016*), in line with a redundant function in some neuronal populations. Redundancy has also been suggested in other cell types (*Shin et al., 2014*). Taken together, the available evidence indicates that the specific requirements of DNM2 functions vary considerably between different cell types and call for further individual examinations.

Previous experiments have shown that DNM2 in SCs is necessary for myelination in dorsal root ganglia explant cultures (*Sidiropoulos et al., 2012*), and it has further been proposed that DNMs may play a role in OL membrane remodeling (*Trajkovic et al., 2006*; *White and Krämer-Albers, 2014*; *Winterstein et al., 2008*). At the functional level, research with mutated versions of CMT-associated *DNM2* in cell culture systems revealed impairments in clathrin-mediated endocytosis (*Sidiropoulos et al., 2012*). However, additional analyses of in vivo models are required to examine the precise function and requirement of DNM2 under physiological and pathological conditions as a prerequisite to thoroughly understand dysfunctions in disease. Thus, we here investigated the requirement for and the functions of DNM2 in myelinating glial cells in appropriate transgenic mice. Specifically, we genetically ablated *Dnm2* both during SC development and in adult SCs. In either case, the mutant mice acquired a peripheral neuropathy with myelination defects. Mechanistically, deletion of *Dnm2* triggered pronounced SC apoptosis, which we have correlated to cytokinesis failure. In contrast, analogous experiments did not reveal a major requirement of DNM2 in OL-mediated myelination.

## Results

### Dynamin 2 in Schwann cells is required for radial sorting of axons and PNS myelination

We performed a systematic in vivo evaluation of the function of DNM2 in SCs defining when and how this function is required. First, we analyzed potential developmental defects in peripheral nerves that lack DNM2 in SCs. To achieve this goal, we carried out loss-of-function experiments by breeding mice containing loxP-flanked *Dnm2* alleles (*Sidiropoulos et al., 2012*) with Myelin Protein Zero (*Mpz*)-Cre mice (*Feltri et al., 1999a*; *Feltri et al., 1999b*) in which the Cre recombinase is expressed in the SC lineage (*Figure 1—figure supplement 1A*). We confirmed by immunoblotting that DNM2 was substantially reduced in postnatal day (P)1 sciatic nerves (SNs) of $Mpz^{Cre}:Dnm2^{loxP/loxP}$ conditional knockout mice (termed P0-Dnm2$^{KO}$) compared to control $Dnm2^{loxP/loxP}$ littermates (*Figure 1A*, *Figure 1—figure supplement 1B*). Residual DNM2 detected in P0-Dnm2$^{KO}$–derived SNs is likely due to its expression in nerve-resident cells other than SCs. Since DNM2 is involved in the fission of vesicles from the plasma membrane during clathrin-mediated endocytosis (CME), we evaluated also the rate of CME with a FACS-based transferrin uptake assay using P1-isolated primary SCs from P0-Dnm2$^{KO}$ and control mice. To allow selection of recombined cells, these animals carried

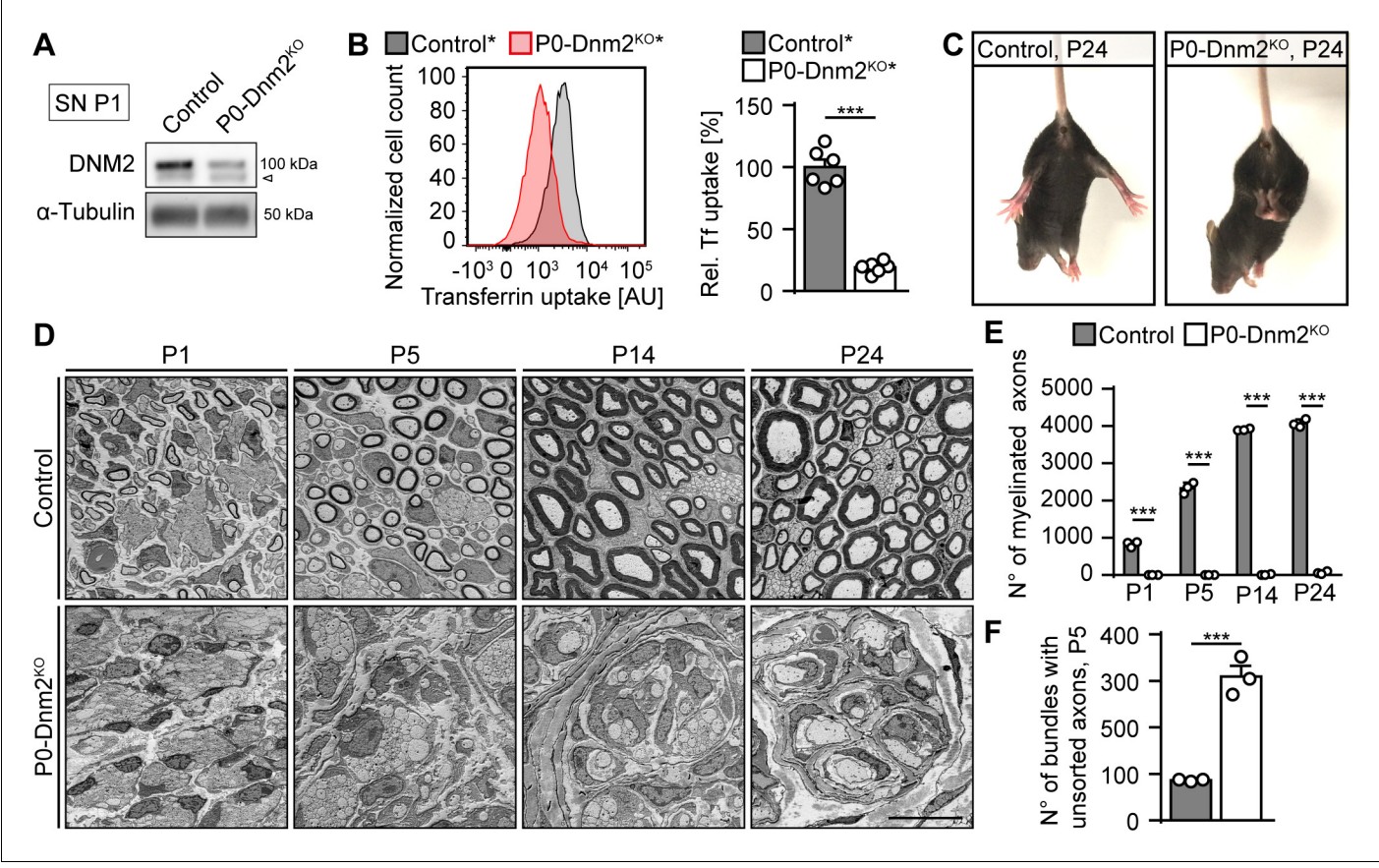

**Figure 1.** Lack of dynamin 2 in Schwann cells results in major radial sorting defects and failure of sciatic nerve myelination. (A) Representative immunoblot of DNM2 in sciatic nerve (SN) extracts derived from *Dnm2^loxP/loxP* (control) and *Mpz^Cre:Dnm2^loxP/loxP* (P0-Dnm2^KO) mice at P1. N = 4 mice/genotype. Arrowhead: unspecific signal. Quantification in *Figure 1—figure supplement 1B*, full-length blots in *Supplementary file 1A*. (B) Representative experimental set (left) and quantified sets (right) of FACS analyses of transferrin (Tf) uptake of YFP+ primary mouse SCs derived from *Mpz^Cre:Rosa26-stop^loxP/loxP-YFP* (control*) and *Mpz^Cre:Dnm2^loxP/loxP:Rosa26-stop^loxP/loxP-YFP* (P0-Dnm2^KO*) P1 SNs. P0-Dnm2^KO* SCs show reduced Tf uptake. Control average was set to 100. N = 6 independent SC preparations/genotype, two-tailed unpaired Student's t-test. (C) Tail-suspension test showing hind paw clasping in P0-Dnm2^KO, but not in control littermate at P24 (consistent phenotype observed in 30 mice/genotype). (D) Exemplary electron microscopy (EM) images showing ultrastructure of control and P0-Dnm2^KO SNs at indicated time points (cross-sections). N = 3 mice/genotype and time point. Scale bar = 10 µm, referring to the entire panel. Quantifications referring to panel D: (E) Number of myelinated fibers/SN cross-section. N = 3 mice/genotype and time point. Two-Way ANOVA with Sidak's multiple comparisons test. (F) Number of bundles with axons > 1 µm/SN cross-section at P5. N = 3 mice/genotype, two-tailed unpaired Student's t-test. Results in graphs represent means ±s.e.m.; ***p<0.001.

DOI: https://doi.org/10.7554/eLife.42404.002

The following figure supplement is available for figure 1:

**Figure supplement 1.** Dynamin 2 ablation in Schwann cells inhibits their differentiation and myelination.
DOI: https://doi.org/10.7554/eLife.42404.003

also a Cre-dependent *Rosa26-YFP* reporter allele (designated P0-Dnm2^KO* and control* to distinguish them from their counterpart animals without the *Rosa26-YFP* reporter) (*Srinivas et al., 2001*). Consistent with loss of DNM2 function, we found a markedly reduced transferrin uptake by *Dnm2*-deficient SCs (*Figure 1B*).

P0-Dnm2^KO mice appeared behaviorally indistinguishable from control littermates up to P10. After P10, a mild shivering-like tremor was evident if mutants were lifted by their tails. By P24, the mutants displayed also pronounced clasping of their hind limbs (*Figure 1C*). We then analyzed the underlying morphological changes in SNs of P0-Dnm2^KO and control littermates at various developmental time points (*Figure 1D*). Already at P1, we found a profound reduction in the numbers of myelinated axons in P0-Dnm2^KO mice compared to control littermates (*Figure 1D,E*). While the quantity of myelinated axons increased from P1 to P24 in control animals as expected, myelinated

fibers remained almost undetectable in P0-Dnm2$^{KO}$ mice (*Figure 1E*). Consistent with these changes, the myelin proteins P0, MAG, and MBP were nearly absent in P1-SNs of P0-Dnm2$^{KO}$ (*Figure 1—figure supplement 1C,D*). We reasoned that the observed lack of myelin might be either the result of early developmental defects, such as faulty radial sorting of large caliber axons by SCs, a prerequisite for myelination (*Feltri et al., 2016*), or of intrinsically defective myelination. To distinguish between these possibilities, we assessed the progression of radial sorting at P5, a time point at which this process is active. We counted the bundles that contained axons with a diameter larger than 1 μm (i.e. definitively destined for sorting and myelination) in SN cross-sections and found a strong increase in P0-Dnm2$^{KO}$ mice compared to controls (*Figure 1F*). These data indicate that developing SNs require DNM2 in SCs for correct radial sorting of axons.

After sorting, SCs undergo transcriptional alterations that mark the onset of myelination. This process includes silencing of negative regulators of myelination such as cJUN and activation of the myelin program that is mainly driven by KROX20 (EGR2). While cJUN is highly expressed by immature SCs and progressively downregulated in pro-myelinating and myelinating SCs, KROX20 shows the opposite expression pattern (*Parkinson et al., 2008*). To evaluate whether loss of DNM2 alters the expression of these markers, we performed immunoblot analyses of P1 SN lysates. These examinations revealed markedly increased amounts of cJUN and reduced KROX20 levels in P0-Dnm2$^{KO}$ mice compared to controls (*Figure 1—figure supplement 1E,F*). A plausible interpretation of these results is that DNM2 is needed for the molecular control of SC differentiation and the onset of myelination, although the changes might partially also reflect the observed delay in radial sorting. Taken together, our data show that DNM2 in SCs is essential for accurate axonal radial sorting and subsequent myelination.

## Dynamin 2 in Schwann cells is necessary for PNS myelin maintenance

The critical function of DNM2 in SC development precluded the examination of the potential role of this protein in SCs of adult nerves. Thus, we generated mice in which conditional *Dnm2* ablation in SCs can be induced after development is completed and myelin has been successfully formed. We crossed mice carrying *loxP*-flanked *Dnm2* alleles (*Sidiropoulos et al., 2012*) with *Mpz$^{CreERT2}$* mice (*Gomez-Sanchez et al., 2017*; *Leone et al., 2003*; *Ribeiro et al., 2013*) (*Figure 2—figure supplement 1A*) and induced recombination at 8 to 10 weeks of age by injecting tamoxifen for five consecutive days. In some experiments, we also included the Cre-dependent *Rosa26-YFP* allele to track the fate of recombined SCs. At 4 weeks post-tamoxifen injection (4 wpT), immunoblotting confirmed that DNM2 levels were reduced in SNs of *Mpz$^{CreERT2}$:Dnm2$^{loxP/loxP}$* mice (P0ERT2-Dnm2$^{KO}$) compared to controls (tamoxifen-injected *Dnm2$^{loxP/loxP}$* mice) (*Figure 2A*, *Figure 2—figure supplement 1B*). Residual DNM2 protein is likely due to non-recombined SCs that account for approximately 30% of total SCs as estimated by quantification of SOX10+ YFP+ recombined SCs (*Figure 3—figure supplement 4B*), together with probable contributions by other DNM2-expressing cells located in SNs.

Induced P0ERT2-Dnm2$^{KO}$ mice were behaviorally indistinguishable from controls up to 4 wpT. After 4 wpT, mutants started to show impaired righting responses, progressing to ataxic gait and finally to a moderate paraparesis at approximately 6 wpT (clinically assessed as described by *King et al. (1983)*) (*Figure 2B,C—source data 1*). Remarkably, all P0ERT2-Dnm2$^{KO}$ mice improved rapidly thereafter, reaching normal motor function again between 7 wpT and 8 wpT (*Figure 2B*). Subsequently, mutant mice remained indistinguishable from control mice until 41 wpT (last time point analyzed). Electrophysiological measurements performed as schematized in *Figure 2G* revealed reduced motor nerve conduction velocities (mNCV) in P0ERT2-Dnm2$^{KO}$ at 6 wpT (*Figure 2H,I*). A severe conduction block was also present in the nerves of these mice as indicated by the almost complete absence of compound motor action potentials (CMAPs) when a proximal stimulus was applied (*Figure 2H*, red profile) and by the strongly reduced ratio between proximal and distal CAMPs (*Figure 2J*). These defects were resolved by 14 wpT and remained at control levels until 41 wpT, indicating that demyelination and subsequent remyelination are likely causes of the progressive clinical phenotype and its recovery in P0ERT2-Dnm2$^{KO}$ mice.

To confirm this interpretation, we examined SN morphology before the onset of symptoms (4 wpT), at the peak of disease (6 wpT), and in the recovery phase (8, 14, 41 wpT). No detectable differences were evident between controls and mutants at 4 wpT (*Figure 2D,E*). However, coinciding with the peak of motor impairment at 6 wpT, SNs of P0ERT2-Dnm2$^{KO}$ contained approximately 15%

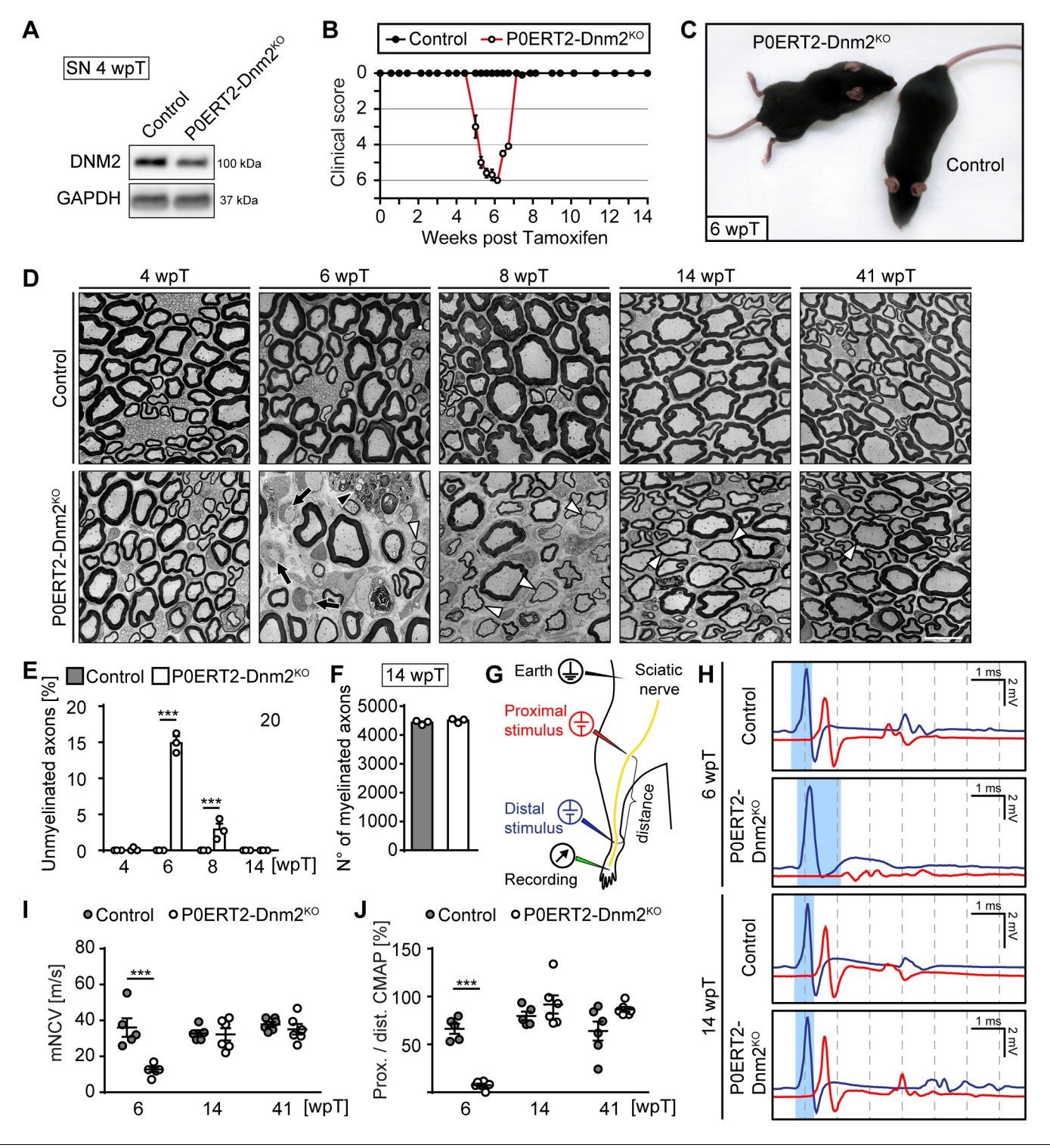

**Figure 2.** Schwann cell-specific ablation of dynamin 2 in adult mice causes a transient neuropathy. (**A**) Immunoblot analysis of DNM2 in SN extracts of *Dnm2^{loxP/loxP}* (control) and *Mpz^{CreERT2}:Dnm2^{loxP/loxP}* (P0ERT-Dnm2^{KO}) mice at 4 weeks post tamoxifen (wpT). N = 3 mice/genotype. Quantification in *Figure 2—figure supplement 1B*, full-length blots in *Supplementary file 1B*. (**B**) Time line of functional impairments and recovery of P0ERT-Dnm2^{KO} mice after tamoxifen treatment based on clinical assessments over 14 weeks. Clinical score: 0 = normal, 1 = less lively, 2 = impaired righting reflex, 3 = absent righting reflex, 4 = ataxic gait, 5 = mild paraparesis, 6 = moderate paraparesis, 7 = severe paraparesis. N = 5 mice/genotype. (**C**) Exemplary picture showing moderate paraparesis of a P0ERT-Dnm2^{KO} mouse, but not of a control littermate at 6 wpT. (**D**) Exemplary EM images showing ultrastructure of control and P0ERT2-Dnm2^{KO} SN cross-sections at the indicated time points. N = 3 mice/time point and genotype. Black arrows:

*Figure 2 continued on next page*

*Figure 2 continued*

Unmyelinated axons; white arrowheads: Thinly melinated axons associated with remyelinating SCs; black arrowhead points to a macrophage. Scale bar = 10 μm, referring to entire panel. (E) Quantification of unmyelinated axons relative to total number of axons with calibers > 1 μm. At least 200 axons/sample were analyzed (random EM fields). N = 3 mice/time point and genotype, Two-Way ANOVA with Sidak's multiple comparisons test. (F) Quantification of myelinated fibers/SN cross-section at 14 wpT. N = 3 mice/genotype, two-tailed unpaired Student's t-test. (G) Schematic drawing of the setup for electrophysiological measurements. (H) Representative traces of electrophysiological findings in control and P0ERT2-Dnm2$^{KO}$ mice SNs at 6 wpT and 14 wpT. At 6 wpT, impulse conduction appears almost completely blocked upon proximal stimulation in P0ERT2-Dnm2$^{KO}$ mice and only a small and dispersed compound muscle action potential (CMAP) was recorded. These deficits were restored at 14 wpT. The blue-shaded columns illustrate the slowing in nerve conduction at 6 wpT in a P0ERT-Dnm2$^{KO}$ mouse and the recovery to normal values at 14 wpT. Blue profile: CMAP upon distal stimulation, red profile: CMAP upon proximal stimulation. (I) Motor nerve conduction velocity (mNCV) in control and P0ERT-Dnm2$^{KO}$ mice at the indicated time points. mNCV is reduced in P0ERT-Dnm2$^{KO}$ mice at 6 wpT, but is restored at 14 wpT. N = 5–6 mice/group, Two-Way ANOVA with Sidak's multiple comparisons test. (J) Conduction block is indicated by severe reduction of the proximal to distal CMAP ratio in P0ERT-Dnm2$^{KO}$ mice at 6 wpT. This is resolved by 14 wpT. N = 5–6 mice/group, Two-Way ANOVA with Sidak's multiple comparisons test. Results in graphs represent means ±s.e.m.; ***$p < 0.001$.

DOI: https://doi.org/10.7554/eLife.42404.004

The following source data and figure supplement are available for figure 2:

**Source data 1.** Phenotypical assessment of control and P0ERT-Dnm2$^{KO}$ mice over a period of 14 weeks after tamoxifen treatment.
DOI: https://doi.org/10.7554/eLife.42404.006
**Figure supplement 1.** Dynamin 2 ablation in adult Schwann cells induces transient cellular dedifferentiation.
DOI: https://doi.org/10.7554/eLife.42404.005

of demyelinated (i.e. myelination-competent but unmyelinated axons) (*Figure 2D,E*, black arrows; 14.91 ± 0.78 in mutant SNs versus none detected in control SNs; mean ±s.e.m.), rarely detected thinly remyelinated axons (*Figure 2D*, white arrowheads), and frequent immune cells such as macrophages (*Figure 2D*, black arrowhead). In parallel to the clinical recovery, demyelinated axons in mutant mice appeared to get progressively remyelinated. Accordingly, at 14 wpT, we found no demyelinated axons (*Figure 2D,E*) and the number of myelinated axons reached control values (*Figure 2F*). As described in other types of de- and remyelination, including in peripheral neuropathies of different types (*Dyck and Thomas, 2005*; *Sanders and Whitteridge, 1946*; *Schröder, 1972*; *Stoll et al., 1986*; *Suter and Scherer, 2003*), myelin sheaths were thinner after remyelination at 41 wpT also in our experimental setting (*Figure 2—figure supplement 1C,D*), combined with shorter internodes as assessed at 14 wpT using teased nerve fiber preparations (*Figure 2—figure supplement 1E*).

In the PNS, demyelination usually involves the functionally important process of SC dedifferentiation (*Jessen et al., 2015*). In this light, we evaluated expression levels of markers for dedifferentiated SCs. By immunohistochemistry, we found an increase of cells that were strongly positive for P75$^{NTR}$ (CD271) in SNs of mutant mice at 6 wpT, followed by a decrease to virtually control levels by 14 wpT (*Figure 2—figure supplement 1F*). Additionally, P75$^{NTR}$ transcript levels were increased in P0ERT2-Dnm2$^{KO}$ SNs compared to controls already at 4 wpT, together with other known molecules that are affected by dedifferentiation such as *Egr1* (Krox24) and cyclin D1 (*Ccnd1*) (*Figure 2—figure supplement 1G*). At 6 wpT, also cJUN protein levels were prominently increased in P0ERT2-Dnm2$^{KO}$ SNs compared to controls and remained elevated up to 14 wpT (*Figure 2—figure supplement 1H, I*). These results confirmed SC dedifferentiation in P0ERT2-Dnm2$^{KO}$ nerves followed by apparent remyelination.

Taken together, our results show that loss of DNM2 in fully differentiated SCs leads to a rapid but reversible demyelination, temporarily severely compromising peripheral nerve function. We conclude that DNM2 in SCs is indispensable for myelin maintenance.

## Dynamin 2 is crucial for Schwann cell survival in both developing and adult peripheral nerves

Based on the observations described above, we hypothesized that common mechanisms may underlie the phenotypes observed in both settings. Given the conserved functions of DNM2 in important cellular processes, we investigated such potentially shared processes by turning to a global approach. We compared the transcriptomes of P0-Dnm2$^{KO}$ mice in developing SNs at P1 with those of P0ERT2-Dnm2$^{KO}$ in SNs at 4 wpT by RNAseq, together with their respective controls. We found

468 transcripts that were jointly downregulated and 1405 transcripts that were jointly upregulated in both P0-Dnm2$^{KO}$ and P0ERT2-Dnm2$^{KO}$ mutants (cutoff: 50% reduction or 50% increase compared to control levels, FDR < 0.05) (*Figure 3A*). Gene ontology analysis revealed that functions of the jointly downregulated genes were related to cholesterol biosynthesis (*Figure 3A—source data 1*), consistent with the reduced myelin content in mutant nerves (*Figures 1* and *2*). The jointly upregulated set was enriched in genes associated with inflammatory responses, such as macrophage recruiting and/or stimulatory signals (*Cd68, Csf1, Cd86, Mif, Cebpa*), interleukins (*Il1b, Il6*), chemokines (*Ccl2, Ccl3*), and inflammatory cell markers (*Ptprc* (CD45)) (*Figure 3A,B*). We also noticed increased levels of transcripts associated with apoptosis (*Figure 3B*), including surface receptors that can mediate apoptotic signals (*Fas, Tnfrsf1a, Tnfrsf10b*), the death receptor adaptor *Fadd*, caspases (*Casp1, Casp3, Casp4, Casp6, Casp8, Casp12*), and caspase activator (*Pycard*). The inclusion of an additional set of transcriptome data from P5 corroborated these findings (*Figure 3B*).

We proceeded with experiments to confirm the results of the RNAseq analysis. First, we analyzed the presence and identity of inflammatory cells in mutant nerves. We detected more CD45+ inflammmatory cells in both P0-Dnm2$^{KO}$ and P0ERT2-Dnm2$^{KO}$ mice compared to controls (*Figure 3C,D*). During development, the number of immune cells was increased at P5 and P14 in P0-Dnm2$^{KO}$ SNs, progressing with age (*Figure 3C*). In SN maintenance, we found the highest number of CD45+ cells at 6 wpT (*Figure 3D*), concomitant with the peaking clinical phenotype and demyelination. Macrophages (CD68+) and T-cells (CD3+) were the prevalent populations of inflammatory cells in P0ERT2-Dnm2$^{KO}$ mice (*Figure 3—figure supplement 1A,B*). Neutrophils and mast cells were also increased, but to a lower extent (*Figure 3—figure supplement 1C,D*). Next, we examined the functional role of macrophages in adult demyelination and remyelination, that is, in P0ERT2-Dnm2$^{KO}$ mice, using clinical recovery as an indirect readout. To do this, we applied clodronate liposome injections at the start of phenotype development (i.e. 4 wpT, *Figure 3—figure supplement 2A*) to cause successful macrophage reduction (*van Rooijen and Hendrikx, 2010*) (*Figure 3—figure supplement 2B,C,D*). This treatment resulted in delayed functional recovery of mutant mice, coherent with a beneficial role of macrophages in the speed of recovery (*Figure 3—figure supplement 2E—source data 1*).

Motivated by our RNAseq results, we next evaluated apoptosis. In developing SNs, immunohistochemistry for cleaved-caspase 3 (cC3+) in combination with the SC marker SOX10, revealed a robust increase in apoptotic SCs in P0-Dnm2$^{KO}$ mice compared to controls at P5 and P14 (*Figure 3E*), in agreement with the transcriptomics data. Analyses of the cellular composition of mutant SNs revealed a complex picture: the total number of SOX10+ SCs per SN cross-section at P5 (*Figure 3—figure supplement 3B*) was substantially reduced compared to controls, despite the fact that the total number of DAPI+ nuclei was unaltered (*Figure 3—figure supplement 3A*), resulting in a decreased cellular proportion of SCs in P5 mutant nerves (*Figure 3—figure supplement 3C*). As expected for controls, the number of SOX10+ SCs per SN cross-section decreased between P5 and P14 (*Figure 3—figure supplement 3B*). In P0-Dnm2$^{KO}$ SNs, however, the number of SOX10+ SCs remained similar between P5 and P14, while the number of total nuclei was robustly increased (*Figure 3—figure supplement 3A*), as was the cross-sectional area of the mutant SNs (*Figure 3—figure supplement 3F*). Infiltration of inflammatory cells is a probable contributor to these effects (*Figure 3C*). Additionally, we found putatively invading GLUT1+ perineurial cells by immunolabeling in mutants (*Figure 3—figure supplement 3D,E*). This finding was confirmed morphologically by EM analysis in P14 P0-Dnm2$^{KO}$ SNs (*Figure 1D* and *Figure 3—figure supplement 3G*). Such perineurial-like cells within the endoneurium have been described previously if SCs were reduced in numbers (*Benninger et al., 2007*; *Grove et al., 2007*; *Jacob et al., 2011*; *Monk et al., 2011*), or when SCs failed to secret DHH to allow correct maturation of the perineurium (*Parmantier et al., 1999*).

In adult SNs, similarly to the developing SNs, we found increased levels of cleaved-caspase 3 protein in P0ERT2-Dnm2$^{KO}$ mice at 6 wpT compared to controls (*Figure 3—figure supplement 4C,D*). These findings indicate elevated SC apoptosis, consistent with the massive loss of recombined SCs observed by staining for the recombination marker YFP in P0ERT2-Dnm2$^{KO*}$ mice (*Figure 3F*, *Figure 3—figure supplement 4B*). Remarkably, very few YFP-positive cells were left at 14 wpT, in parallel with the observed recovery of the clinical phenotype. These results indicate that upon loss of DNM2, recombined SCs undergo apoptosis, while non-recombined cells do not and are capable of remyelinating the sciatic nerve with high efficiency. Consistent with this interpretation, DNM2 expression was reduced in recombined YFP-positive SCs at 4 wpT, while at 14 wpT, together with

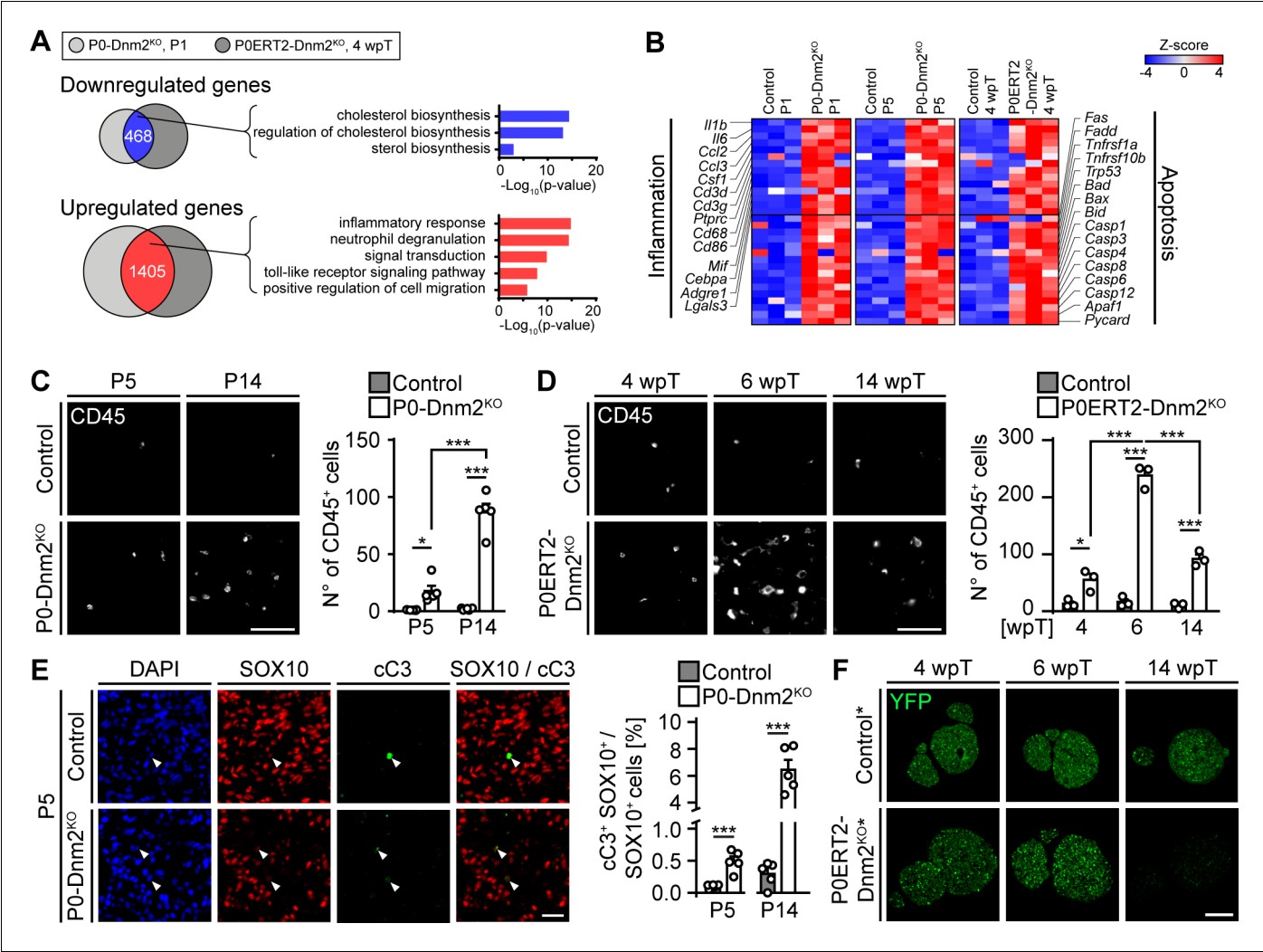

**Figure 3.** Mice lacking dynamin 2 in Schwann cells display characteristics of inflammation and Schwann cell death. (**A**) Gene ontology analysis of jointly downregulated or upregulated genes in P1 P0-Dnm2KO and 4 wpT P0ERT2-Dnm2KO SNs compared to controls. (**B**) Heat map showing representative differentially expressed genes and their GO categories in controls versus Dnm2KO SNs or P0CreERT2-Dnm2KO SNs at the indicated time points. N = 3 mice/genotype. (**C**) (Left) Immunostainings of controls and P0-Dnm2KO SN cross-sections for the inflammatory cell marker CD45 at P5 and P14. (Right) Quantification of CD45+ cells/SN cross-sections. N = 5 mice/time point and genotype. Two-Way ANOVA with Holm-Sidak's multiple comparisons test. Scale bar = 50 μm for entire panel. (**D**) (Left) Immunostainings of controls and P0ERT2-Dnm2KO SN cross-sections for the inflammatory cells marker CD45 at 4 wpT, 6 wpT, and 14 wpT. (Right) Quantification of CD45+ cells/SN cross-sections. N = 3 mice/time point and genotype. Two-Way ANOVA with Holm-Sidak's multiple comparisons test. Scale bar = 50 μm for entire panel. (**E**) (Left) Immunostainings of controls and P0-Dnm2KO SN cross-sections for cleaved-caspase 3 (cC3) in combination with the SC marker SOX10 at P5. (Right) Percentage of cC3+ SOX10+ SCs at P5 and P14. N = 5 mice/time point and genotype, two-tailed unpaired Student's t-test. Scale bar = 25 μm for entire panel. (**F**) Immunostainings for YFP (recombined cells) on SN cross-sections derived from *Mpz^{CreERT2}:Rosa26-stop^{loxP/loxP}-YFP* (control*) and *Mpz^{CreERT2}:Dnm2^{loxP/loxP}:Rosa26-stop^{loxP/loxP}-YFP* (P0ERT2-Dnm2^{KO}*) mice at 4 wpT, 6 wpT, and 14 wpT. YPF+ cells were no longer detectable at 14 wpT in P0ERT2-Dnm2KO mice. N = 3 mice/time point and genotype. Scale bar = 200 μm for the entire panel. Results in graphs represent means ±s.e.m. *p<0.05, **p<0.01, ***p<0.001.

DOI: https://doi.org/10.7554/eLife.42404.007

The following source data and figure supplements are available for figure 3:

**Source data 1.** Transcriptomes of sciatic nerves from P0-Dnm2KO and control mice at P1 and P5, and from P0ERT2-Dnm2KO and control mice at 4 wpT.
DOI: https://doi.org/10.7554/eLife.42404.013
**Figure supplement 1.** Various immune cells invade sciatic nerves after dynamin 2 depletion in adult Schwann cells.
DOI: https://doi.org/10.7554/eLife.42404.008
**Figure supplement 2.** Macrophages are beneficial for recovery after demyelination due to loss of dynamin 2 in adult Schwann cells.
DOI: https://doi.org/10.7554/eLife.42404.009

*Figure 3 continued on next page*

*Figure 3 continued*

**Figure supplement 2—Source data 1.** Phenotypical assessment of control and P0ERT-Dnm2[KO] mice treated with Clodronate or Saline from day 27 to day 64 post tamoxifen.
DOI: https://doi.org/10.7554/eLife.42404.010
**Figure supplement 3.** Lack of dynamin 2 in Schwann cells of developing nerves causes Schwann cell loss and invasion of perineurial cells.
DOI: https://doi.org/10.7554/eLife.42404.011
**Figure supplement 4.** Adult dynamin 2-depleted Schwann cells are replaced by dynamin 2-positive Schwann cells in remyelinated nerves.
DOI: https://doi.org/10.7554/eLife.42404.012

the loss of YFP-positive cells, SC expression of DNM2 was restored (*Figure 3—figure supplement 4A*).

Taken together, these results show that both developing and adult SCs depend critically on DNM2 for their survival and that, upon DNM2 loss, cells other than SCs invade developing peripheral nerves. Furthermore, our results indicate a remarkable plasticity of adult peripheral nerves allowing functional recovery after substantial SC loss.

## Dynamin 2 is required for proper cell cycle progression, mitotic entry, and cytokinesis of Schwann cells

Radial sorting defects and decreased SC numbers as seen in P0-Dnm2[KO] mice are frequently associated with reduced SC survival, reduced proliferation, or a combination of both. The high rates of SC apoptosis indicated that impaired radial sorting in P0-Dnm2[KO] mice might be due to the reduced SC survival. To determine whether altered proliferation also contributes to the initial findings and to analyze the cell cycle more closely, we assessed SC proliferation by 5-ethynyl-2'-deoxyuridine (EdU) incorporation in combination with Ki-67 immunostaining. As schematically represented in *Figure 4A*, EdU labels cells during S phase, while Ki-67 labels proliferating cells irrespective of the cell cycle phase. We found that the percentages of both EdU+ or Ki-67+ SCs (SOX10+) were substantially increased in P0-Dnm2[KO]sciatic nerves compared to controls at P5 (*Figure 4B,C,D*), indicating higher SC proliferation in mutants. However, further analysis revealed that also the proportion of EdU + among the Ki-67+ SCs was increased in P0-Dnm2[KO] compared to controls at P5 (*Figure 4B,E*), suggesting abnormal cell cycle progression. Since DNM2 has been mechanistically implicated in progression through the M phase and, in particular, in cytokinesis (*Conery et al., 2010*; *Smith and Chircop, 2012*; *Thompson et al., 2002*), we hypothesized that *Dnm2*[KO] SCs may have defects in entering mitosis and/or during mitotic progression. To investigate these aspects, live-cell imaging was carried out on YFP+ SCs isolated from P0-Dnm2[KO]\* and control\* mice at P1. We found a decrease in SC mitotic events in *Dnm2*[KO] compared to control SCs (*Figure 4F*), associated with an increase in mitosis duration in *Dnm2*[KO] SCs (*Figure 4G,J*). Additionally, mutant cells were more often multinucleated compared to controls, at both 48 and 72 hr after isolation (*Figure 4H,I*), likely as a result of impaired cytokinesis. Indeed, we could document failures in cytokinesis in mutant cells (*Figure 4K*).

Collectively, our results indicate that, together with ensuring SC survival, DNM2 in SCs is necessary to ensure proper cell cycle progression, mitotic entry, and cytokinesis.

## Dynamin 2 is not essential for oligodendrocyte differentiation and CNS myelination

Given the central roles of DNM2 in SC development and myelination, we wondered whether also oligodendrocytes (OLs), the myelinating glia of the CNS, are similarly dependent on DNM2 function. OL lineage cells express DNM1, DNM2 and DNM3 (*Marques et al., 2016*; *Zhang et al., 2014*), and cell culture studies have suggested that dynamins regulate OL differentiation and myelin protein localization upon axonal stimulation (*Kippert et al., 2007*; *Simons and Trotter, 2007*; *Trajkovic et al., 2006*; *White and Krämer-Albers, 2014*). DNM2 is expressed throughout the OL lineage (*Zhang et al., 2014*) and thus we asked whether DNM2 is required for OL differentiation and myelination in vivo. To answer this question, we generated mice with conditional *Dnm2* ablation in the OL lineage by crossing mice carrying *loxP*-flanked *Dnm2* (*Dnm2[loxP]*) alleles (*Sidiropoulos et al., 2012*) with *Cnp[Cre]* mice (*Genoud et al., 2002*; *Lappe-Siefke et al., 2003*) (*Figure 5—figure supplement 1A*). Behaviorally, *Dnm2* conditional knockout mice (CNP-Dnm2[KO]) were

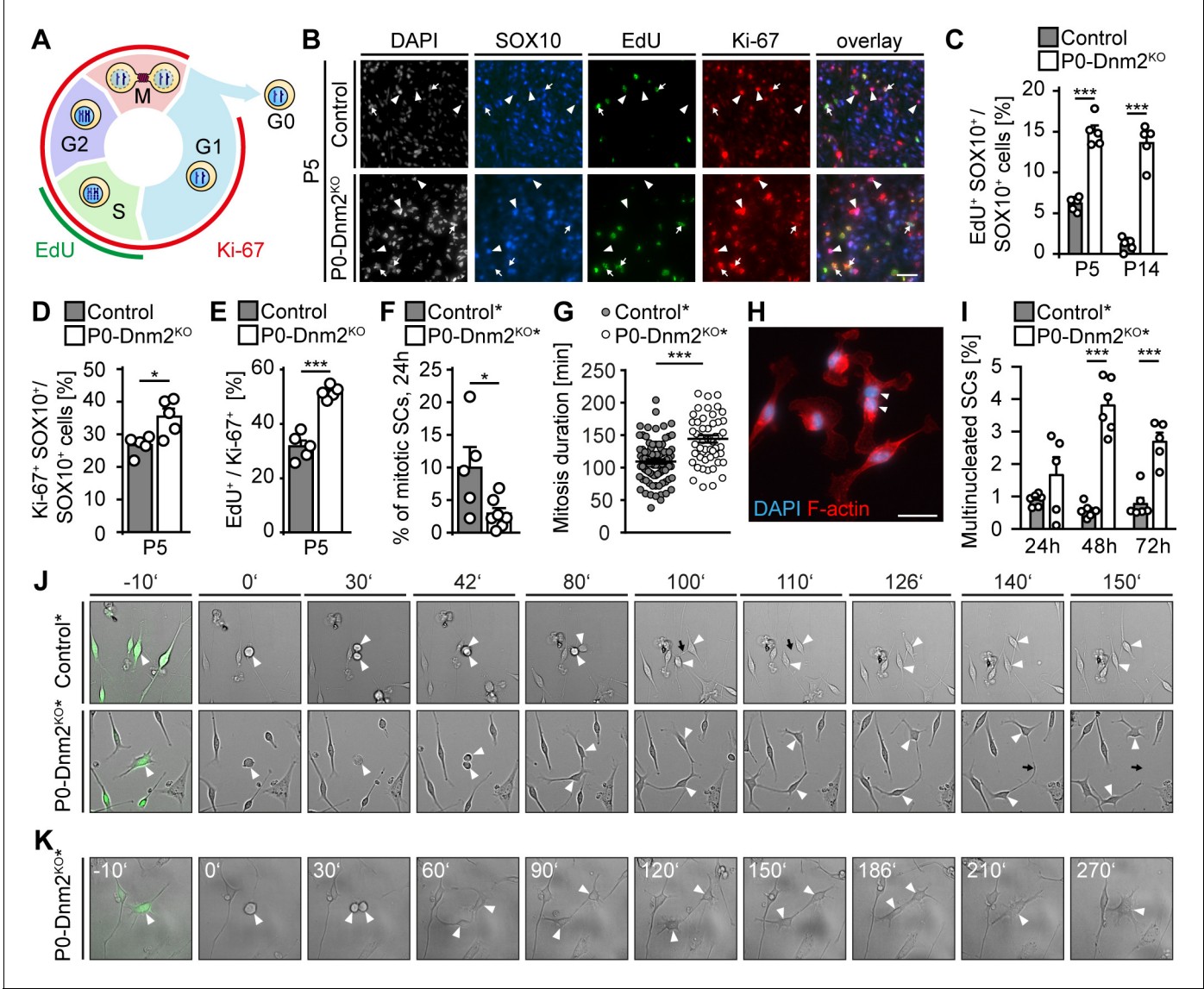

**Figure 4.** Schwann cells lacking dynamin 2 show impaired cell cycle progression, reduced mitosis rate, and cytokinesis defects. (A) Schematic representation of the cell cycle phases marked by Ki-67 and EdU. (B) EdU-labeling, combined with immunostainings for Ki-67 and SOX10 on control and P0-Dnm2^KO SN cross-sections at P5. Arrows: EdU+ Ki-67+ SOX10+ SCs, arrowheads: Ki-67+ SOX10+ SCs. Scale bar = 25 μm for entire panel. (C) Quantification of (B). Percentage of EdU+ SCs/SN cross-sections. N = 5 mice/time point and genotype, Two-Way ANOVA with Sidak's multiple comparisons test. (D) Quantification of (B). Percentage of Ki-67+ SCs/SN cross-sections at P5. N = 5 mice/genotype, two-tailed unpaired Student´s t-test. (E) Quantification of (B). Percentage of EdU+ among Ki-67+ SCs/SN cross-section at P5. N = 5 mice/genotype, two-tailed unpaired Student´s t-test. (F) Quantification of mitotic events in cultured mouse SCs isolated from *Mpz^Cre:Rosa26-stop^loxP/loxP-YFP* (control*) and *Mpz^Cre:Dnm2^loxP/loxP: Rosa26-stop^loxP/loxP-YFP* (P0-Dnm2^KO*) SNs at P1, monitored by time-lapse microscopy for 24 hr. Each data point represents one individual animal (at least 42 cells/animal analyzed). N = 5 controls and seven mutant mice, two-tailed unpaired Student´s t-test. (G) Quantification of mitosis duration (minutes) of SCs of control* and P0-Dnm2^KO* mice, monitored by time-lapse microscopy for 24 hr. Each data point represents one cell derived from a total of 5 control and seven mutant mice. Cells derived from each animal were isolated and analyzed separately, but pooled in one graph; two-tailed unpaired Student´s t-test. (H) Exemplary picture of a multinucleated Dnm2^KO SCs after 48 hr in culture. Scale bar = 25 μm. (I) Quantification of multinucleated cells in control and Dnm2^KO SCs after 24 hr, 48 hr and 76 hr in culture. N = 6 mice/genotype for 48 hr; N = 6 control* and n = 5 P0-Dnm2^KO* mice for 24 hr and 72 hr, two-Way ANOVA with Sidak's multiple comparisons test. (J) Exemplary time-lapse images of the mitosis of control and Dnm2^KO SCs (YFP+ cells). Arrowheads: cell body of dividing SCs, black arrows: cytokinesis site. (K) Representative time-lapse images of Dnm2^KO SC (YFP+) failing cytokinesis. The SC is undergoing mitosis, but fails to divide (arrowheads point to the cell body). Results in graphs represent means ±s.e.m.; *p<0.05, ***p<0.001.

DOI: https://doi.org/10.7554/eLife.42404.014

similar to P0-Dnm2$^{KO}$, indistinguishable from control littermates up to P10, but displaying pronounced clasping of the hind limbs when lifted by their tails by P24. PNS and CNS deficiencies may contribute to these observations, since CNP-Cre drives recombination in both SCs and OLs. We focused our further analysis of these animals on CNS tissues. First, we determined by immunoblotting that DNM2 was effectively reduced in spinal cords from CNP-Dnm2$^{KO}$ compared to control littermates (P14, P60) (*Figure 5A*, *Figure 5—figure supplement 1B*). To confirm depletion of DNM2 in mature OLs, we co-stained P14 spinal cord sections from CNP-Dnm2$^{KO}$ and control mice for DNM2 and the marker CC1. As expected and contrary to controls, mutant CC1+ OLs showed no detectable specific staining for DNM2 (*Figure 5B*). Analogous to the previous experiments with DNM2-deficient SCs, we then assessed the effect of loss of DNM2 function by measuring transferrin uptake using primary OL lineage cells isolated from brain of control* and CNP-Dnm2$^{KO}$* mice (carrying also the Cre-dependent *Rosa26-YFP* reporter allele) at P7 and cultured for 14 days. In contrast to our findings in SCs, transferrin uptake was not detectably reduced in *Dnm2*$^{KO}$ OL lineage cells (*Figure 5C*), despite an effective decrease of DNM2 levels as determined by immunoblotting (*Figure 5—figure supplement 1C*).

We then assessed the impact of DNM2 loss on OL development and differentiation by analyzing OL lineage cell numbers and differentiation markers in multiple CNS regions at P14, including spinal cord, corpus callosum, and cerebellar white matter lobes IV-V. No significant differences were found between control and CNP-Dnm2$^{KO}$ mice in the number of total OL lineage cells (OLIG2+), oligodendrocyte progenitor cells (OPCs; PDGFRα+ OLIG2+), or mature OLs (CC1+ OLIG2+) in all regions analyzed (*Figure 5D,E*, *Figure 5—figure supplement 1D*). We further examined CNS myelination in CNP-Dnm2$^{KO}$ and control littermates by electron microscopy (*Figure 5F*). No detectable differences were found in the number of myelinated axons or in myelin thickness (g-ratio) in the spinal cord ventral white matter (P14, P60) and the corpus callosum (P60) (*Figure 5F,G*, *Figure 5—figure supplement 1E,F*).

Overall our observations indicate that DNM2 is largely dispensable for OL differentiation and CNS myelination, in contrast to the essential role of DNM2 in SCs. Compensatory functional mechanisms, potentially including other isoforms of dynamins, may contribute to these findings. However, we can also not exclude subtle effects of lack of DNM2 on OL differentiation (or on important earlier functions of DNM2 in OL lineage development) that we may not have detected due to potential experimental limitations. These considerations include our selection of the CNP Cre-driver mouse line, although the same line has been used successfully for such analyses before (*Bercury et al., 2014*; *Genoud et al., 2002*; *Lebrun-Julien et al., 2014*; *Wahl et al., 2014*).

## Discussion

In this study, we demonstrate that developing SCs require the specific functions of the large GTPase DNM2 for radial sorting of large caliber axons and their subsequent myelination. In addition, we show that DNM2 in SCs is also essential for myelin maintenance. Using loxP/Cre recombination-mediated gene ablation in mice, we found that lack of DNM2 in developing or adult SCs resulted in a severe peripheral neuropathy, characterized by SC loss and signs of inflammation. Removal of DNM2 specifically from adult SCs induced fast demyelination, coupled with SC dedifferentiation and ensued by widespread apoptosis of recombined DNM2-deficient SCs. Remarkably, this striking cell loss was rapidly compensated for by a minor fraction of non-recombined DNM2-expressing SCs, paralleled by complete functional recovery of the animals within two weeks after the peak of symptoms. At the cellular level, our data indicate that DNM2 is crucial for SC survival and differentiation, both in development and in adulthood, and acts as a prominent regulator of the cell cycle. The intimate dependency of SCs on DNM2 is consistent with a particular susceptibility of these cells to neuropathy-associated *DNM2* mutations. Triggered by the observed severe consequences of DNM2 deficiency in peripheral myelinating glia, we examined also whether DNM2 is necessary for OL in CNS myelination. However, we found no evidence supporting this hypothesis.

Radial sorting is the process in which SCs contact and segregate axons destined for myelination (i.e. axons with diameters larger than 1 μm). The requirements for successful radial sorting include dynamic cytoskeletal rearrangements, mechanisms of axonal recognition and selection, and progressive expansion of the SC population to match the number of axons (*Feltri et al., 2016*). To achieve correct numbers, SCs in developing nerves are subject to a fine balance between proliferation and

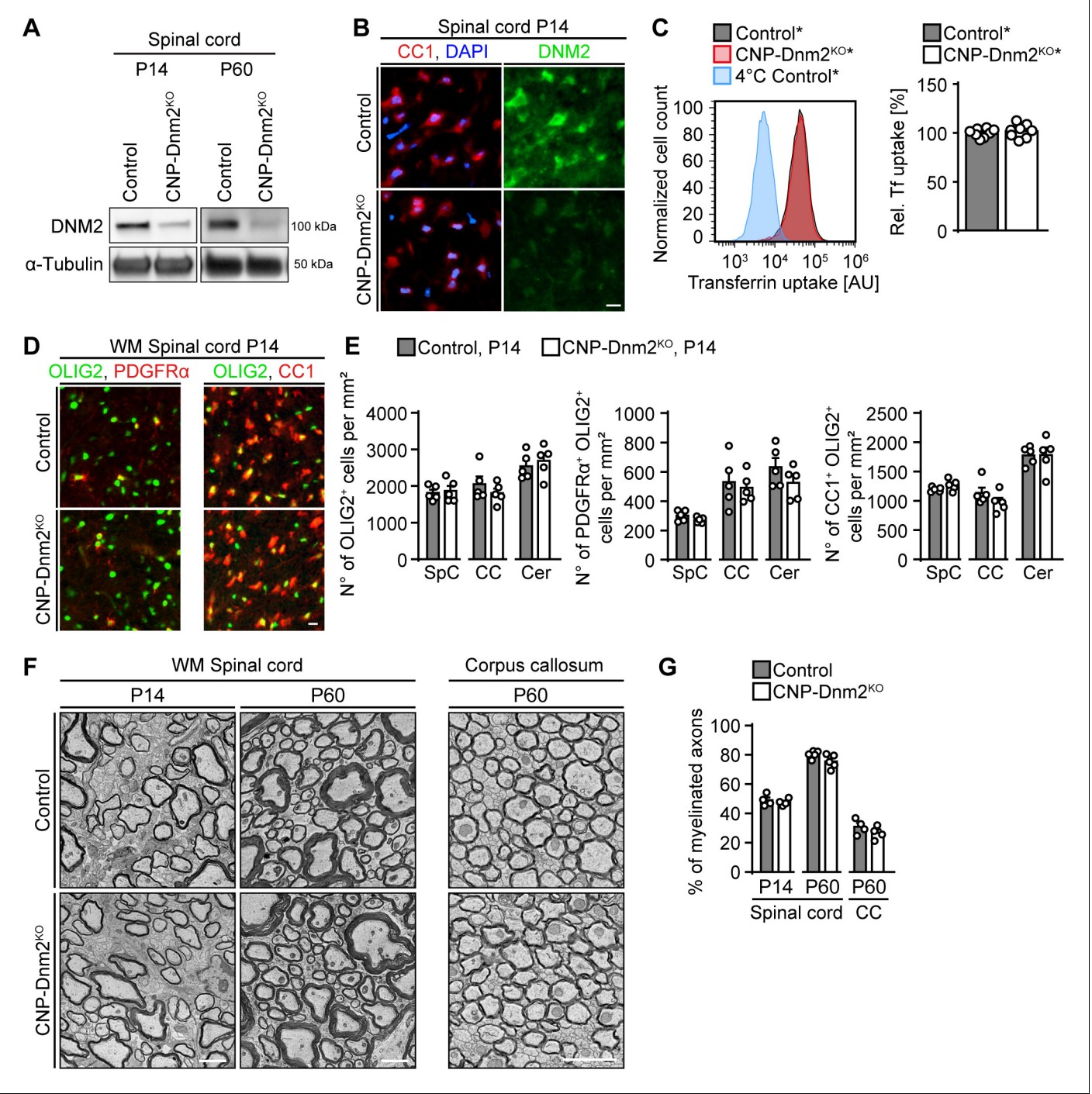

**Figure 5.** Oligodendrocytes can differentiate and myelinate without dynamin 2. (**A**) Immunoblot analysis of DNM2 in spinal cord extracts from *Dnm2^{loxP/loxP}* (control) and *Cnp^{Cre}: Dnm2^{loxP/loxP}* (CNP-Dnm2^{KO}) mice (P14, P60). N = 4 mice/genotype. Quantification in *Figure 5—figure supplement 1B*, full-length blot in *Supplementary file 1C*. (**B**) Exemplary transverse sections of ventral horn region of lumbar spinal cord from control and CNP-Dnm2^{KO} mice at P14, double-stained for CC1 (red; mature oligodendrocyte (OL) marker) and DNM2 (green), showing DNM2 ablation in mutant OLs. N = 3 mice/genotype. Scale bar = 10 μm. (**C**) Representative experimental set (left) and quantified sets (right) of FACS analyses of transferrin (Tf) uptake on primary mouse OLs derived from *Cnp^{Cre}:Rosa26-stop^{loxP/loxP}-YFP* (control*) and *Cnp^{Cre}:Dnm2^{loxP/loxP}:Rosa26-stop^{loxP/loxP}-YFP* (CNP-Dnm2^{KO}*) P7 brain (14 days in culture). CNP-Dnm2^{KO} OLs show no detectable difference in Tf uptake compared to controls (note the virtually complete overlap of the corresponding peaks). Control average is set to 100. N = 8 mice/genotype, two-tailed unpaired Student's t-test. (**D**) Representative immunostainings of control and CNP-Dnm2^{KO} in white matter (WM) ventral spinal cord cross-sections for OLIG2 and PDGFRα (left panel) or OLIG2 and CC1 (right panel) at P14. N = 5 mice/genotype. Scale bar = 10 μm. (**E**) Quantification of (**D**) (Spinal cord (SpC) ventrolateral WM hemisection, corpus

*Figure 5 continued on next page*

*Figure 5 continued*

callosum (CC) midbody, and cerebellum (Cer) lobes IV,V). Total OL lineage cells (OLIG2+), OPC (PDGFRα+ OLIG2+) and mature OL (CC1+ OLIG2+) in SpC ventrolateral WM, corpus callosum (CC), and cerebellum (Cer) of control and CNP-Dnm2$^{KO}$ mice. N = 5 mice/genotype, two-tailed unpaired Student's t-test. (F) Exemplary EM images showing ultrastructure of control and CNP-Dnm2$^{KO}$ ventral SpC WM (P14, P60) and CC (P60). Scale bars = 2 μm. (G) Quantification of (F) (random fields). Percentages of myelinated fibers in SpC ventral WM (P14, P60) and corpus callosum (P60). At least 752 axons/animal were analyzed. N = 4–5 control and mutant mice at the indicated time points, two-tailed unpaired Student's t-test. Results in graphs represent means ±s.e.m.

DOI: https://doi.org/10.7554/eLife.42404.015

The following figure supplement is available for figure 5:

**Figure supplement 1.** Dynamin 2-depleted oligodendrocytes do not show detectable defects in differentiation and myelination.

DOI: https://doi.org/10.7554/eLife.42404.016

apoptosis (*Jessen and Mirsky, 2005*). Reduced SC proliferation is correlated with incomplete radial sorting in mouse models (*Benninger et al., 2007*; *Berti et al., 2011*; *Brinkmann et al., 2008*; *Grove et al., 2007*; *Porrello et al., 2014*), and excessive apoptosis can lead to a similar outcome (*Jacob et al., 2011*; *Messing et al., 1992*). In some cases, dual regulation of proliferation and apoptosis allows cells to reach a new equilibrium that remains compatible with proper sorting (*D'Antonio et al., 2006*). Increased SC proliferation and survival can instead accelerate radial sorting (*Figlia et al., 2017*; *Grigoryan et al., 2013*). We found that after DNM2 deletion in developing SCs, mutant SNs contained fewer SCs compared to controls at P5, concomitant with increased apoptosis and impaired radial sorting. This increased apoptosis could explain the reduced number of SCs, especially if not counteracted by higher levels of proliferation. However, parallel analysis using the proliferation marker Ki-67 and EdU incorporation revealed increased numbers of labelled SCs compared to controls. These results were surprising and thus we asked why the apparently increased proliferation was not able to counteract apoptosis to restore appropriate SC number in P0-Dnm2$^{KO}$ nerves. Subsequent experiments revealed that *Dnm2*$^{KO}$ SCs are affected at various stages of the cell cycle, including 1) impaired entry into the mitotic phase, 2) longer duration of the mitotic process until complete cell division, which translated into 3) fewer mitotic events completed in the same time frame as in control SCs. Furthermore, *Dnm2*$^{KO}$ SCs were more prone to cytokinesis failure compared to controls. DNM2 has been previously implicated in cytokinesis (*Smith and Chircop, 2012*), and our evidence is coherent with such a role also in SCs. It is likely that cytokinesis failure, with the consequent formation of multinucleated cells, contributes to apoptosis and ultimately the decrease in SC numbers in P0-Dnm2$^{KO}$ mice at P5, analogous to observations made in other cell types (*Chircop et al., 2011*; *Conery et al., 2010*; *Joshi et al., 2011*; *Redgrove et al., 2016*).

Besides appropriate cell number expansion, successful radial sorting is dependent on coordinated SC cytoskeletal rearrangements in order to extend processes that reach axons and establish stable interaction required for the following myelination step. Alterations in stability and dynamic arrangements of actin filaments or microtubules affect SC process extension and radial sorting (*Grove et al., 2007*; *Sparrow et al., 2012*) and/or myelination (*Jin et al., 2011*; *Novak et al., 2011*). Since DNM2 is involved in regulating actin and microtubule dynamics and stability (*Gu et al., 2010*; *Tanabe and Takei, 2009*; *Yamada et al., 2016*), potential cytoskeleton abnormalities in P0-Dnm2$^{KO}$ SCs may also partially contribute to the impaired radial sorting phenotype.

Once an axon segment is sorted from the bundle, the associated pro-myelinating SC undergoes extensive transcriptional changes, generally characterized by shutdown of proliferation-related (myelin-suppressing) transcripts, and upregulation of the myelination program chiefly driven by key transcription factor KROX20 (*Decker et al., 2006*; *Topilko et al., 1994*). Thus, the processes of SC radial sorting, proliferation and ensuing differentiation/myelination are closely linked. Our analysis revealed almost complete absence of myelin in P0-Dnm2$^{KO}$ nerves, also on rare axons successfully sorted from bundles. These observations prompted further investigations of markers differentially regulated at the onset of SC differentiation/myelination. We found strongly increased levels of cJUN coupled to near complete absence of KROX20 in P0-Dnm2$^{KO}$ nerves. Such imbalances on the expression of opposing differentiation regulators can negatively influence SC differentiation (*Parkinson et al., 2008*), consistent with the interpretation of impaired SC differentiation at the onset of myelination as an additional consequence of DNM2 deficiency.

DNM2 is widely associated with clathrin-mediated endocytosis which we found to be strongly reduced in DNM2$^{KO}$ SCs. Thus, DNM2-based control mechanisms of SC radial sorting and differentiation may also include regulation of endocytosis and recycling of cell surface proteins such as ErbB2/ErbB3-, integrin β1-, and GPR126-receptors. These proteins have been previously implicated in SC radial sorting (*Feltri et al., 2016*) and can be internalized via DNM2-dependent endocytosis (*Paul et al., 2015*; *Sorkin and Goh, 2009a*; *Sorkin and von Zastrow, 2009b*). Furthermore, an impact of DNM2-mediated internalization on surface receptors (potentially also altering signaling) is indirectly supported by studies in cell culture, showing that expression of a neuropathy-associated DNM2 mutant alters surface levels of integrin β1 and ErbB2 (*Sidiropoulos et al., 2012*). The individual contributions of such potentially altered signaling pathways to the observed phenotypes caused by DNM2 deficiency in development and in the adult remain to be elucidated. With regard to disease mechanisms of neuropathy-causing dominantly inherited *DNM2* mutations, generation and careful analysis of animal models carrying such mutations are required.

In adult nerves, SCs remain strictly dependent on the continuous action of transcriptional regulators to preserve myelin maintenance (*Bremer et al., 2011*; *Brügger et al., 2015*; *Decker et al., 2006*). We found that also DNM2 function is essential in adult SCs to maintain myelination. Induced loss of DNM2 specifically in adult SCs evoked an acute and robust demyelination, accompanied by clinical neuropathy and coupled with SC dedifferentiation and invasion of inflammatory cells. SC dedifferentiation was followed by progressive loss of the recombined DNM2-negative SC pool (i.e. approx. 70% of all SCs originally present). Despite the prominent pathology developed in these nerves and the widespread loss of SCs, there was robust histological and functional recovery. This recovery was mediated by SCs that did not lose DNM2 in our experimental system (i.e. approx. 30% of the original SC pool). Thus, our results demonstrate an astonishing ability of adult mouse peripheral nerves to repair damage if faced by massive SC death and demyelination. These findings may also have significant overtones for our understanding of human nerve diseases since demyelination and dedifferentiation of SCs, coupled with low grade invasion of inflammatory cells, are common features of acquired and inherited peripheral neuropathies (*Martini and Willison, 2016*). One might speculate that if such prominent reparative mechanisms as observed in our model are sufficiently effective, secondary axonal damage might be limited or prevented allowing efficient recovery. Thus, fostering the intrinsic regenerative capability of SCs might be a promising therapeutic target in acquired neuropathies and/or in hereditary neuropathies, including in some forms of CMT where this process might be chronically reduced due to genetic mutations.

Of note, we cannot determine currently to which extent the observed regenerative capability in our mutants may include the participation/recruitment of non-myelinating Schwann cells that are normally associated with Remak bundles (*Murinson et al., 2005*). To answer this conceptually important question requires the development of reliable tools that allow specific tracing of such cells (*Ma et al., 2018*). Similarly, we cannot exclude contributions of progenitors turning into SCs in our experimental setting (*Petersen and Adameyko, 2017*; *Stierli et al., 2018*).

Myelination in the PNS and CNS encompasses numerous similarities and yet striking differences, including the cell types that perform this function, the structure and composition of the myelin they produce, and the molecular pathways that govern myelination and homeostasis of the axon-myelin unit. The striking reliance of SCs on DNM2 motivated us to investigate whether DNM2 is also necessary for OL myelination. Quite unexpectedly, our data indicate that DNM2 is not essential for OL myelination in our in vivo experimental model. Previous in vitro experiments suggested that in OLs, dynamins contribute to myelin protein re-localization, allowing plasma membrane specialization and possibly affecting myelination. These experiments used overexpression of dominant-negative forms of DNM2, which may have caused some unintended gain-of-function effects, or chemical antagonists which may not have been completely specific (*Trajkovic et al., 2006*; *Winterstein et al., 2008*). It appears likely that unexplored compensatory mechanisms, possibly including other DNM isoforms, substitute for DNM2 functions in OL in vivo.

In conclusion, our study shows that DNM2 in SCs is essential for proper peripheral nerve development and myelin maintenance. The key findings highlight that SCs are dependent on DNM2 for clathrin-mediated endocytosis, cell cycle progression and cytokinesis, differentiation, myelination, and maintenance of the myelinated state in adult mice. In contrast, specific DNM2 functions in OL are largely dispensable for CNS myelination, in line with the PNS-specific defects of CMT-causing DNM2 mutations. Elimination of DNM2 in SCs of adult mice caused widespread SC death. Remarkably,

these experiments revealed an astonishing self-healing capability of peripheral nerves when affected by massive SC death and demyelination. Fostering the underlying highly efficient regenerative process is an appealing target for future therapeutic approaches in selected peripheral nerve diseases.

## Materials and methods

### Experimental animals

To achieve conditional deletion of *Dnm2* in Schwann cells, *Dnm2^{loxP/loxP}* mice (Dnm2 <tm1Ueli>) (*Sidiropoulos et al., 2012*) were crossed with mice carrying a *Cre* transgene under the control of the *Mpz* promoter (Tg(Mpz-cre)26Mes/J; RRID:IMSR_JAX:017927) (*Feltri et al., 1999a*). These breedings gave rise to *Mpz^{Cre}:Dnm2^{loxP/loxP}* (referred to as P0-Dnm2^{KO}) and to *Dnm2^{loxP/loxP}* (referred to as control) mice.

To achieve inducible conditional deletion of *Dnm2* in adult Schwann cells, *Dnm2^{loxP/loxP}* were crossed with *Mpz^{CreERT2}* (Tg(Mpz-cre/ERT2)2Ueli) mice (*Leone et al., 2003*). These breedings gave rise to *Mpz^{CreERT2}:Dnm2^{loxP/loxP}* (referred to as P0ERT2-Dnm2^{KO}) and *Dnm2^{loxP/loxP}* (referred to as control) mice. At 8–10 weeks of age these mice were intraperitoneally injected with 2 mg of tamoxifen (Sigma, #T5648) once a day on five consecutive days to induce recombination. A 20 mg/mL tamoxifen solution was prepared dissolving the powder in 10% ethanol/Sunflower Seed Oil (Sigma, #S5007).

To achieve conditional deletion of *Dnm2* in oligodendrocytes, *Dnm2^{loxP/loxP}* mice were crossed with *Cnp^{Cre}* (Cnp^{tm1(cre)KAN}) mice (*Genoud et al., 2002*; *Lappe-Siefke et al., 2003*). These breedings gave rise to *Cnp^{Cre}:Dnm2^{loxP/loxP}* (referred to as CNP-Dnm2^{KO}), and *Dnm2^{loxP/loxP}* (referred to as control) mice.

For some experiments, labelling of recombined cells was required. Therefore, the *Rosa26-stop^{loxP/loxP}-YFP* mouse line (GtROSA26Sor < tm1(EYFP)Cos>; RRID:IMSR_JAX:006148) (*Srinivas et al., 2001*) was used. These mice express a cytosolic YFP upon Cre-induced recombination. In such experiments, *Mpz^{CreERT2}:Rosa26-stop^{loxP/loxP}-YFP* mice were used as controls, referred to as control*, and *Mpz^{CreERT2}:Dnm2^{loxP/loxP}:Rosa26-stop^{loxP/loxP}-YFP* mice are referred to as P0ERT2-Dnm2^{KO}*. The same type of nomenclature was used for the *Mpz^{Cre}* and *Cnp^{Cre}* lines.

Genotypes were determined through genomic PCR using the following primers:

Cre: forward 5′-ATCGCCAGGCGTTTTCTGAGCATAC-3′ reverse 5′-GCCAGATTACGTATATCCTGGCAGC-3′

Dnm2: forward 5′-GGGAATCCTGCTGGGGAAGCTCTC-3′ reverse 5′-CTCTAGCACTTCCACTAAGCCCTCC-3′

RosaYFP:

Primer-1 5′-AAAGTCGCTCTGAGTTGTTAT-3′

Primer-2 5′-GCGAAGAGTTTGTCCTCAACC-3′

Primer-3 5′-GGAGCGGGAGAAATGGATATG-3′

Mice of either sex were used in the experiments. P0-Dnm2^{KO}, CNP-Dnm2^{KO}, and respective control mice were on a C57B6/J background. P0ERT2-Dnm2^{KO} and control mice were backcrossed at least six times in a C57B6/J background. Mice were housed with a maximum number of five animals/cage, kept in a 12 hr light-dark cycle, and fed standard chow *ad libitum*. Note: The severity of the disability of P0-Dnm2^{KO} and CNP-Dnm2^{KO} mice increased steadily over 60 days, and the animals could not be kept longer on welfare grounds. All animal experiments were performed with the approval and in strict accordance to the guidelines of the Zurich Cantonal Veterinary Office under permits ZH129/2011, ZH161/2014, and ZH090/2017.

### Clodronate administration

Starting at 4 weeks post-tamoxifen (4 wpT), control and P0ERT2-Dnm2^{KO} mice were intravenously injected with 100 µl of clodronate liposomes (5 mg/ml) in phosphate buffer (LIPOSOMA BV www.clodronateliposomes.com) or with saline. Every mouse received a tail vein injection every third day for a total of three treatments (4 wpT, 4 wpT +3 d, 4 wpT +6 d).

## Electrophysiology

Motor nerve conduction was measured in anaesthetized mice as described (*Zielasek et al., 1996*) with a modification in anesthesia (ketamine (Ketanest[R]70 µg/g mouse) and xylazine (Rompun[R]7 µg/g mouse) using a Neurosoft Evidence 3102 electromyograph (Schreiber and Tholen Medizintechnik, Germany) as described (*Krieger et al., 2014*; *Niemann et al., 2014*). The examiner was blinded as to the experimental allocation of mice. In brief, upon supramaximal stimulation of the tibial nerve at the ankle ('distal') and of the sciatic nerve (SN) at the sciatic notch ('proximal'), the elicited compound motor nerve action potentials (CMAPs) were recorded with a pair of steel needle electrodes in the foot muscles. Supramaximal indicates stimulation at least 30% above the electric square wave direct current needed to obtain a maximal CMAP. Throughout the recordings, limb temperature was maintained at 32–34°C. Nerve conduction velocities were calculated in m/s from distal and proximal latencies. In addition, 10 successive F-waves were recorded and the shortest latencies were taken upon stimulation at the ankle. Ratios of the proximal and distal CMAPs were calculated to assess conduction block along the sciatic nerve.

## Electron microscopy (EM)

SNs were dissected and fixed with 3% glutaraldehyde and 4% paraformaldehyde in 0.1 M phosphate buffer. To isolate spinal cords and brains, mice were first perfused with 3% glutaraldehyde and 4% paraformaldehyde in 0.1 M phosphate buffer. After dissection SNs, spinal cords, and brains were treated with 2% osmium tetroxide, dehydrated over a series of acetone gradients (30%, 50%, 70%, 90%, 96%, and 100%), and embedded in Spurrs resin (Electron Microscopy Sciences). Ultrathin sections (99 nm) from SNs from P0Cre-Dnm2[KO] and control mice were collected on ITO coverslips (Optics Balzers) and the entire cross-section was imaged with either a Zeiss Gemini Leo 1530 FEG or a Zeiss Merlin FEG scanning electron microscopes attached to ATLAS modules (Zeiss). Ultrathin sections (65 nm) of SNs from P0ERT2-Dnm2[KO] and control mice were collected on copper grids (Electron Microscopy Sciences; G200-Cu) and random fields were imaged with a FEI Morgagni 268 TEM. Ultrathin sections (99 nm) of lumbar spinal cord and corpus callosum from CNP-DNM2 and control mice were collected on ITO coverslips (Optics Balzers). The ventral funiculus of the white matter and the corpus callosum body above the hippocampal region were imaged with Zeiss Merlin FEG scanning electron microscope attached to ATLAS modules (Zeiss).

## Morphological analysis and g-ratio measurements

Quantification of the number of myelinated axons, myelin abnormalities, and number of bundles with unsorted axons in P0Cre-Dnm2[KO] and control mice were performed on entire SN cross sections. Quantification of the percentage of unmyelinated axons (unmyelinated axons relative to the total number of axons analyzed >1 µm) in P0ERT2-Dnm2[KO] and control sciatic nerves was performed on random field images. At least 200 axons/sample were analyzed. Quantification of the percentage of myelinated axons (myelinated axons relative to the total number of axons analyzed) in CNP-Dnm2[KO] and control spinal cords and corpora callosa were performed on three selected areas of 900 and 225 µm$^2$ / section, respectively. At least 752 axons/sample were analyzed.

To calculate the g-ratio, the axon diameter was derived from the axon area, while the fiber diameter was calculated by adding to the axon diameter twice the average of the myelin thickness measured at two different locations of the myelin ring. For CNP-Dnm2[KO] and control spinal cords and corpora callosa, three random fields were selected from the panorama images and at least 108 fibers/animal were analyzed. For the P0ERT2-Dnm2[KO] and control sciatic nerves at least 90 fibers/animal were analyzed in the random field images acquired.

## Teased osmicated nerve fibers

Immediately after dissection, SNs were fixed with 3% glutaraldehyde and 4% paraformaldehyde in 0.1 M phosphate buffer. SNs were further treated with 1% osmium tetroxide (Electron Microscopy Sciences) for 1 hr, followed by a series of glycerol gradients (30%, 60% and 100%) at 55°C for 12 hr each. Samples were teased in 100% glycerol and imaged using an epifluorescence microscope (Zeiss Axio Imager.M2) equipped with a monochromatic CCD camera (sCMOS, pco.edge). Quantification of at least 100 internodes per animal was performed in ImageJ (version 1.50i).

## Antibodies and chemicals

The following primary antibodies were used: DNM2 (Pineda, 1:100; or GeneTex, #GRX109652, RRID:AB_1950134, 1:200), cJUN (CST, #9165, RRID:AB_2130165, 1:1000), MAG (Invitrogen, # 34–6200, RRID:AB_2533179, 1:1000), P0 (Millipore, #ab9352, RRID:AB_571090, 1:1000), MBP (Serotec, MCA409S, RRID:AB_325004, 1:200), cleaved caspase 3 (CST, #9664, RRID:AB_2070042, 1:1000) α-tubulin (Sigma-Aldrich, T5168, RRID:AB_477579, 1:1000), GAPDH (Hytest, 5G4 6C5, RRID:AB_1616722, 1:10000), SOX10 (R and D Systems, AF2864, RRID:AB_442208, 1:200), GFP/YFP (Aves Lab, #GFP-1020, RRID:AB_10000240, 1:1000), CD45 (BD, #550539, RRID:AB_2174426, 1:200), Ki-67 (abcam, #ab15580, RRID:AB_443209, 1:200), CC1 (Calbiochem, OP80, RRID:AB_2057371, 1:500), OLIG2 (Millipore, #MABN50, RRID:AB_10807410, 1:1000 or Millipore, #AB9610, RRID:AB_570666, 1:500), PDGFRα (CST, #3174, RRID:AB_2162345, 1:600), p75$^{NTR}$ (Millipore, #ab1554, RRID:AB_11211656, 1:200), CD68 (Serotec, #MCA1957, RRID:AB_322219, 1:100), CD3 (Dako, #A0452, RRID:AB_2335677, 1:200), GR1 (BD, #553127, RRID:AB_394643, 1:200), CD117 (BD, #562417, RRID:AB_11154233, 1:200), GLUT-1 (Millipore, #07–1401, RRID:AB_1587074, 1:200). Primary antibodies against KROX20 (1:1000) were a generous gift from Dr. Dies Meijer (Edinburgh). Alexa488-coupled phalloidin (Life Technologies #A12379, used 1:100). HRP- and fluorophore-conjugated secondary antibodies were purchased from Jackson ImmunoResearch and used 1:800 for immunostainings or 1:5000 for western blots.

## Immunostainings

SNs were dissected, fixed for one hour in 4% paraformaldehyde (PFA), cryopreserved in 20% sucrose overnight, and then embedded in optimal cutting temperature medium OCT (TissueTek). Spinal cords and brains were isolated after mice have been intracardially perfused with PBS followed by 4% PFA, postfixed overnight in 4% PFA 5% sucrose, washed in PBS, dehydrated in 30% sucrose overnight, embedded in OCT (TissueTek), and stored at −80°C.

Cryosections (10 μm thick) were cut with an increment of 100 μm between serial sections, transferred to SuperFrost Plus (Thermo Scientific) coated slides, and stored at −80°C until further use. Slides with frozen sections were thawed at room temperature for 30 min, rehydrated with 1x PBS for 10 min, permeabilized with 0.5% triton X-100, blocked with blocking buffer (1% BSA, 10% goat or donkey serum, 0.1% triton X-100 in PBS) for one hour, incubated overnight with primary antibodies, incubated for one hour with fluorophore-conjugated secondary antibodies, counterstained with DAPI (Life Technologies), and mounted with Vectashield (Vector Laboratories). For spinal cord and brain, an alkaline antigen retrieval procedure was added between the rehydration and the blocking steps: slides were immersed in sodium alkaline buffer (40 mM Trizma-Base, 0.75 mM EDTA, pH9), placed in a water bath at 86°C for 20 min, cooled down at room temperature for 20 min, and washed twice in PBS. Cell cultures were fixed with 4% PFA for 15 min followed by the same staining procedure.

For cell proliferation experiments, mouse pups were injected with 50 μg EdU (Thermo Fisher Scientific)/gram of body weight and sacrificed one hour later. EdU staining was performed using the Click-iT EdU Alexa488 kit (Thermo Fisher Scientific, #C10337) as per manufacturer's instructions.

Immunostainings were imaged using an epifluorescence microscope (Zeiss Axio Imager.M2) equipped with a monochromatic CCD camera (sCMOS, pco.edge). Three representative sections/sample were imaged and analyzed for immunostainings involving P0-Dnm2$^{KO}$ and CNP-Dnm2$^{KO}$, while one representative section was imaged and analyzed for immunostainings involving P0ERT2-Dnm2$^{KO}$ mice.

For control and CNP-Dnm2$^{KO}$ spinal cords, three hemi-sections/animal were quantified and averaged. The distinction between grey and white matter was based on MBP staining. For corpora callosa, three sections/animal were quantified in the body region and averaged. For cerebella, three sections/animal were quantified in lobes V-VI and averaged.

## Western blots

Immediately after dissection, SNs were placed in ice-cold PBS, the epineurium and perineurium were removed as much as possible, and the nerves were snap-frozen in liquid nitrogen and stored at −80°C until further processing. Spinal cords were dissected and the meninges were removed before being snap-frozen in liquid nitrogen. To prepare lysates, SNs or spinal cords were ground on dry ice,

mixed with PN2 lysis buffer (25 mM Tris-HCl pH7.4, 95 mM NaCl, 10 mM EDTA, 2% SDS, 1% prote-ase inhibitor (PIC Sigma P8340) and phosphatase inhibitors (Sigma P5726 and P0044)), boiled for 5 min, and centrifuged at 17000 g for 10 min at 16°C. 10 μg of proteins/sample were mixed 1:4 with sample buffer (200 mM Tris-HCl pH6.8, 40% glycerol, 8% SDS, 20% β-mercaptoethanol, 0.4% bro-mophenol blue), boiled for 5 min, run on 4–15% polyacrylamide gradient gels (Biorad), and blotted onto a PVDF membrane (Millipore, #IPVH00010). After blocking with 5% BSA in PBS with Tween 0.1% (PBS-T), membranes were incubated overnight with primary antibodies diluted in 1% BSA in PBS-T. Chemiluminescent signals were generated using HRP-conjugated secondary antibodies and ECL (GE Healthcare) or ECL prime (GE Healthcare), and were detected using Fusion FX7 (Vilber Lourmat). Quantification of band intensities was performed with ImageJ (version 1.50i). α-tubulin or GAPDH were used as normalization controls. The molecular size shown next to the cropped bands refers to the apparent molecular weight according to Precision Plus Protein Standards (#161–0373, BioRad).

## Preparation and culture of primary neonatal Schwann cells (SCs)

SNs were dissected from P1 P0-Dnc2[KO]* and control* mouse pups, the epineurium and perineurium were removed as much as possible, and the nerves were digested with 1.25 mg/ml trypsin (Sigma-Aldrich, #T9201) and 2 mg/ml collagenase (Sigma-Aldrich, #C0130) for one hour at 37°C. Cells were then centrifuged, resuspended in D-medium (DMEM-Glutamax plus 10% FCS (Life Technologies)), and 150 × 10³ cells were seeded on 3,5 cm PLL-coated plates for transferrin uptake assays, or 14,000 cells were seeded on laminin-coated 8-well chambered coverglass (Nunc Lab-Tek, 155411) for live imaging, while 30,000 cells were seeded on laminin-coated 1.2 cm coverslips for immunos-tainings. The cultures obtained contained typically about 80% SCs and 20% fibroblasts, with SCs dis-tinguished by the presence of YFP as recombination/lineage marker.

## Preparation and culture of primary oligodendrocyte lineage cells

OL lineage cells were isolated from P7 brains of control* and CNP-Dnm2[KO]* mice as described in detail by *Emery and Dugas (2013)*. We modified the composition of the SATO growth medium by adding 2% B-27 complement (GIBCO, #17504044), 10 ng/ml FGF (Peprotech, #100-18B), 20 ng/ml PDGFα (Peprotech, #100-13A), and 10 ng/ml CNTF (Peprotech, #450–13). OLs were expanded for 2 weeks, then 100 × 10³ cells were seeded on 3,5 cm PDL-coated plates for follow-up experiments. The cultures obtained contained 80–95% oligodendrocytes and 5–20% astrocytes/microglia, depending on the preparation. Recombined OLs were distinguished by the YFP recombination/line-age marker.

## Transferrin uptake assay

SCs were plated as described above, cultured overnight, and serum starved in DMEM GlutaMAX for 30 min at 37°C. OL were re-plated for two days before the assay was performed. Cells were incu-bated with 10 μg/ml Alexa Fluor 568- or 647-labeled transferrin (Thermo Fisher Scientific, #T23365, #T23366) for 3 min at 37°C to allow endocytosis. As a negative control, we used cells treated in the same way, but kept at +4°C to block transferrin uptake. Subsequently, cells were transferred to ice to stop transferrin uptake, washed twice with cold PBS, acidic stripped (0.2 M Na2HPO4, 0,1 M citric acid) for two minutes to remove non-internalized transferrin, followed by two washes with PBS. Cells were then trypsinized, centrifuged, and resuspended in 400 μl flow buffer (PBS, 2% BSA, 5 mM EGTA) and kept on ice until analyzed. Samples were analyzed on a LSR Fortessa (BD Biosciences, Franklin Lakes, NJ, USA) or a SONY SH800 using about 10 × 10⁴ cells/sample. Data were analyzed with the FlowJo software (RRID:SCR_008520, version 10.0.7). SCs or OL were first gated according to forward and side scatter, followed by gating for the recombination/lineage marker YFP and mea-surement of the internalized transferrin. Between 5000 and 10,000 cells were recorded. To account for differences in signal intensity for each sample, the flow cytometry profiles were normalized.

## Live imaging

14,000 SCs, isolated from P1 control* and P0-Dnm2[KO]* SNs, were seeded on laminin-coated 8-chambered coverglass (Nunc Lab-Tek, 155411) for 24 hr in DMEM (GIBCO) supplemented with 10% FCS (Life Technologies), 2 ng/ml neuregulin 1 (R and D, #396-HB-050) and 2 μM Forskolin

(Calbiochem, #344270). Cells were imaged with a Zeiss 200M or an Olympus IX81 inverted microscope, and a time-lapse series was acquired using a fully motorized stage, X10 objective, and MetaMorph software (version 7.7.11, Molecular Devices) by using the time-lapse modules. Temperature control was achieved by using an incubator box, providing a humidified atmosphere with 5% $CO_2$. Imaging was carried out for 24 hr with a lapse time of 2 min for phase contrast and 6 min for YFP detection. A total of 5 controls* and 7 P0-Dnm2$^{KO}$* animals were analyzed. Cells obtained from each animal were kept separately, and each animal considered as one n. To calculate mitosis duration and percentage of mitotic SCs, movies were quantified by an investigator blinded as to the sample identities.

## RNA extraction for RNA sequencing or qRT-PCR analysis

Immediately after dissection, SNs were placed in ice-cold PBS, the epineurium and perineurium were removed as much as possible, and nerves were snap-frozen in liquid nitrogen and stored at −80°C until needed.

Total RNA was extracted using RNeasy kit (Qiagen) for developmental samples or Qiazol (Qiagen) for adult samples as per manufacturer's instructions. Samples were used for RNA sequencing (see below) or processed for qRT-PCR analysis as follows: 50–200 ng of total RNA were reverse transcribed using Maxima First Strand cDNA Synthesis Kit (Thermo Fisher Scientific) as per manufacturer's instructions. qPCR reactions were performed using FastStart Essential DNA Green Master (Roche) and Light Cycler 480 II (Roche). Sequences of the primers used: P75 forward 5'-CCCCACCAGAGGGAGAGAA-3', reverse 5'-GGCTACTGTAGAGGTTGCCATCA-3'; cyclin D1 forward 5'-TGTTCGTGGCCTCTAAGATGAAG-3', reverse 5'-AGGTTCCACTTGAGCTTGTTCAC-3'; Krox24/Egr1 forward 5'-CAGCGCCTTCAATCCTCAAG-3', reverse 5'-AGCGGCCAGTATAGGTGATG-3'; GAPDH forward 5'-GGTGAAGGTCGGTGTGAACGGATTTGG-3', reverse 5'-GGTCAATGAAGGGGTCG TTGATGGCAAC-3'; β-actin forward 5'-GTCCACACCCGCCACC-3', reverse 5'-GGCCTCGTCACC-CACATAG-3'. Relative mRNA fold changes were obtained by using the $2^{-\Delta\Delta Ct}$ method after normalization to GAPDH or β-actin.

## Illumina RNA-sequencing

### Library preparation

The quality of isolated RNA was determined with a Qubit (1.0) Fluorometer (Life Technologies, California, USA) and a Bioanalyzer 2100 (Agilent, Waldbronn, Germany). Only those samples with a 260 nm/280 nm ratio between 1.8–2.1 and a 28S/18S ratio within 1.5–2 were further processed. The TruSeq RNA Sample Prep Kit v2 (Illumina, Inc, California, USA) was used in the succeeding steps. Briefly, total RNA samples (100 ng) were poly A enriched and then reverse-transcribed into double-stranded cDNA. The cDNA samples were fragmented, end-repaired and polyadenylated before ligation of TruSeq adapters containing the index for multiplexing Fragments containing TruSeq adapters on both ends were selectively enriched with PCR. The quality and quantity of the enriched libraries were validated using Qubit (1.0) Fluorometer and the Caliper GX LabChip GX (Caliper Life Sciences, Inc., USA). The product is a smear with an average fragment size of approximately 260 bp. Libraries were normalized to 10 nM in Tris-Cl 10 mM, pH8.5 with 0.1% Tween 20.

### Cluster Generation and Sequencing

The TruSeq PE Cluster Kit HS4000 or TruSeq SR Cluster Kit HS4000 (Illumina, Inc, California, USA) was used for cluster generation using 10 pM of pooled normalized libraries on the cBOT. Sequencing were performed on the Illumina HiSeq 2500 paired end at 2 × 101 bp or single end 100 bp using the TruSeq SBS Kit HS4000 (Illumina, Inc, California, USA).

The raw reads were first cleaned by removing adapter sequences, trimming low quality ends (four bases from read start and read end), and filtering reads with low quality (phred quality <20) using Trimmomatic (Bolger et al., 2014). Sequence alignment and isoform expression quantification of the resulting high-quality reads to the mouse genome assembly (build GRCm38) was performed with the RSEM algorithm (Li and Dewey, 2011) (version 1.2.22) with the option for estimation of the read start position distribution turned on. Genes not present (<10 counts/gene) in at least 50% of samples from one condition were discarded from further analyses. Differential gene expression analysis was performed using the R/bioconductor package edgeR (Robinson et al., 2010) in which the

normalization factor was calculated by trimmed mean of M values (TMM) method. P-values were adjusted for multiple testing using the Benjamini-Hochberg procedure (*Benjamini and Hochberg, 1995*). Genes showing altered expression (fold change <0.5 or>1.5) with adjusted (Benjamini and Hochberg method) p-value<0.05 (indicated as false discovery rate, FDR) were considered differentially expressed. Within this set of genes, downregulated and upregulated genes were separately subjected to gene ontology analysis of biological processes using the online tool Enrichr (http:// amp.pharm.mssm.edu/Enrichr/).

## Statistical analysis

Data processing and statistical analyses were carried out using GraphPad Prism (version 7.0a) and Microsoft Excel. Data distribution was assumed to be normal and variances were assumed to be equal, although this was not formally tested due to low n number. Sample sizes were chosen according to sample sizes generally employed in the research field. Investigators were blinded to the genotypes during analysis of morphological and immunohistochemical data. No randomization methods were used. Two-tailed unpaired Student's t-test was used if only two conditions or genotypes were compared. In all other cases, one- or two-way ANOVAs followed by Tukey's, Holm-Sidak's, or Sidak's multiple comparisons tests were employed, as indicated in the figure legends and in *Supplementary file 2*. p<0.05 was considered to be statistically significant. No samples or data were omitted during the analyses.

## Data availability

RNA-sequencing data have been deposited in the GEO database under accession number GSE113106.

## Acknowledgements

We thank all members of the Suter lab for discussions, the Functional Genomics Center Zurich and the ScopeM imaging facility of ETH Zürich for excellent technical support, Drs. Klaus-Armin Nave, Maria Laura Feltri, and Lawrence Wrabetz for transgenic mice, and Dr. Dies Meijer for antibodies.

## Additional information

### Funding

| Funder | Author |
| --- | --- |
| Schweizerischer Nationalfonds zur Förderung der Wissenschaftlichen Forschung | Ueli Suter |

The funders had no role in study design, data collection and interpretation, or the decision to submit the work for publication.

### Author contributions

Daniel Gerber, Monica Ghidinelli, Conceptualization, Data curation, Formal analysis, Investigation, Visualization, Methodology, Writing—original draft, Writing—review and editing; Elisa Tinelli, Christian Somandin, Jorge A Pereira, Klaus V Toyka, Conceptualization, Data curation, Formal analysis, Investigation, Methodology, Writing—review and editing; Joanne Gerber, Carsten Wessig, Data curation, Formal analysis, Investigation, Methodology, Writing—review and editing; Andrea Ommer, Conceptualization, Formal analysis, Investigation, Methodology, Writing—review and editing; Gianluca Figlia, Michaela Miehe, Conceptualization, Investigation, Methodology, Writing—review and editing; Lukas G Nägeli, Vanessa Suter, Data curation, Formal analysis, Investigation, Writing—review and editing; Valentina Tadini, Páris NM Sidiropoulos, Investigation, Methodology, Writing—review and editing; Ueli Suter, Conceptualization, Resources, Data curation, Formal analysis, Supervision, Funding acquisition, Project administration, Writing—review and editing

## Author ORCIDs
Monica Ghidinelli http://orcid.org/0000-0002-7378-6273
Andrea Ommer http://orcid.org/0000-0003-1517-7266
Gianluca Figlia http://orcid.org/0000-0001-8689-8488
Ueli Suter http://orcid.org/0000-0002-9211-5184

## Ethics

Animal experimentation: Mice were kept at 12 hr light/dark cycle. All experiments were approved by the Veterinary Office of the Canton of Zurich, Switzerland (reference ZH129/2011, ZH161/2014, and ZH090/2017).

## Decision letter and Author response

Decision letter https://doi.org/10.7554/eLife.42404.023
Author response https://doi.org/10.7554/eLife.42404.024

## Additional files

### Supplementary files

• Supplementary file 1. Full-length western blot images.
DOI: https://doi.org/10.7554/eLife.42404.017

• Supplementary file 2. Statistic summary.
DOI: https://doi.org/10.7554/eLife.42404.018

• Transparent reporting form
DOI: https://doi.org/10.7554/eLife.42404.019

### Data availability

All data generated or analyzed during this study are included in the manuscript and supporting files. Source data files have been provided for Figure 2B; Figure 3 A,B; Figure 3-Figure supplement 2E. RNA-sequencing data have been deposited in the GEO database under accession number GSE113106.

The following dataset was generated:

| Author(s) | Year | Dataset title | Dataset URL | Database and Identifier |
|---|---|---|---|---|
| Gerber D, Ghidinelli M | 2018 | RNA sequencing of control and dnm2-knockout mouse sciatic nerves | https://www.ncbi.nlm.nih.gov/geo/query/acc.cgi?acc=GSE113106 | NCBI Gene Expression Omnibus, GSE113106 |

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
