## [Decision Letter]

Thank you for submitting your article ""Schwann Cells, But Not Oligodendrocytes, Depend Strictly on Dynamin 2 Function" for consideration by *eLife*. Your article has been reviewed by three peer reviewers and the evaluation has been overseen by a Reviewing Editor and Gary Westbrook as the Senior Editor. The reviewers have opted to remain anonymous.

The reviewers have discussed the reviews with one another and the Reviewing Editor has drafted this decision to help you prepare a revised submission.

*Reviewer #1 (General assessment and major comments):*

This thorough and exceptionally well-written manuscript by Gerber et al. describes an analysis of dynamin 2 (DNM2) in the development and maintenance of peripheral and central myelin. The authors use constitutive and inducible knockout strategies to delete DMN2 from Schwann cells (SCs) during development and in adults. Using a broad array of techniques, ranging from electron microscopy to in vivo compound action potential recording, the authors demonstrate that DMN2 is required for both the survival and maintenance of SCs, and thus function of the nerve. Deletion of DMN2 from only a subset of SCs, resulted in rapid recovery, presumably through de-differentiation of unrecombined SCs. Surprisingly, DMN2 was dispensable for oligodendrocyte development, as no deficiencies were noted in CNS myelin. Together, the studies provide a very convincing picture of the selective importance of DMN2 in SCs and the ability of peripheral nerves to recover from transient loss of SCs. The conclusions are somewhat limited, as they do not clearly establish why developing and adult SCs are dependent on DMN2 and why oligodendrocytes are not. In particular, the use of bulk tissue for RNAseq makes it difficult to define the SC specific changes. Nevertheless, the in vivo nature of the analysis provides an important contribution to the field and clarifies the role of this protein in different myelinating cells. I have several minor comments which could improve the manuscript.

1) The studies would be significantly enhanced if the authors were to provide more mechanistic information about why SCs are preferentially dependent on DMN2. Single cell RNAseq of SNs or further interrogation of DMN expression by oligodendrocytes would help in this regard.

2) The authors conclude that apoptotic loss of SCs after deletion of DMN2 results in recovery due to de-differentiation of non-recombined SCs. This is supported by the permanent disappearance of YFP-labeled SCs in the nerve. It would be nice if the authors provided additional evidence that the SCs that appear after partial depletion were produced through de-differentiation of mature SCs (rather than possible mobilization of progenitors) using in vivo fate tracing.

3) In Figure 2B what is the purpose of the shaded color in the graph? A scale should be included.

4) Figure 3—figure supplement 4B should be included in the main figure to accompany panel Figure 3F.

*Reviewer #2:*

This is an interesting, well-documented and important paper using a variety of transgenic mice with specific cut out of dynamin 2 using the loxP system in Schwann cells (P0-Dnm2^KO^) or in both Schwann cells and oligodendrocytes, although data are only shown for oligodendrocytes (CNP-Dnm2^KO^). The laboratory has extensive experience in the use of these mice. The experiments reveal a crucial role for the large GTPase dynamin2 in controlling membrane dynamics in development of Schwann cells. SCs are dependent on DNM2 for clathrin-mediated endocytosis, a role well-established in other cell types and also in cell cycle progression and cytokinesis, Dynamin 2 has important roles both during radial sorting and subsequently in myelination. Radial sorting is severely delayed in Schwann cells lacking dynamin2. There is a failure of Schwann cells to insert themselves within the bundles of axons sequestered in 'families' and at p24, the age at which the animals died, no myelination appears to have occurred in the P0-Dnm2^KO^ animals. Although some Schwann cells appear to have progressed to a 1:1 relationship with axons at this age, they fail to envelop them properly. Levels of myelin proteins are remarkably reduced and negative regulators of myelination such as cJun are elevated as shown in western blots SCs in DNM2 knockout mice. Invasion of the perineurium into the endoneurial space is also clearly visible at p24. Interestingly no such important role was found for dynamin 2 in oligodendrocyte development (CNP-Dnm2^KO^).

Another series of experiments using tamoxifen induced cut out of dynamin2 in adult nerves showed an intriguing result. This cut out induced a transient neuropathy that was maximal at 14 days after the last injection, followed remarkably by slow remyelination and recovery. The authors provide evidence that Schwann cells with dynamin2 cut out die and are replaced by dynamin2 positive Schwann cells present in the nerves. Two papers suggest that use of this P0ERT-Dnm construct cuts out specifically in myelin Schwann cells in adult mice (Ribeiro et al., 2013; Gomez-Sanchez et al., 2017.) adding strength to the authors' suggestion that the remyelination is caused by migration of Remak Schwann cells that fail to die to the demyelinated nerves. There is also evidence for this in older papers from John Griffin's lab, e.g. Murinson, Archer and Griffin, 2005.

In summary an excellent paper, fully worthy of publication in *eLife*.

*Reviewer #3:*

Gerber et al. deleted Dnm2 in SCs (either embryonically or conditionally in young adults). Embryonic deletion results in in the failure of SC maturation, including the radial sorting of axons, with corresponding reduced levels of myelin-related proteins. Dnm2-null SCs failed to endocytose transferrin. Conditionally deleting Dnm2 in young adult mice resulted in progressive motoric impairment for ~6 weeks, followed by improvement. Electrophysiological studies of affected mice showed dispersion and conduction block, implicating demyelination as the mechanism, and this was directly demonstrated in histological studies. The interesting observation here is that the remyelinating SCs were likely derived from non-recombined SCs; the recombined SCs having died by apoptosis. Analysis of cultured SCs show that Dnm2-null SCs have impaired cytokinesis. RNA seq of "P0-Dnm2^KO^ mice in developing SNs at P1 with those of P0ERT2-Dnm2^KO^ in SNs at 4 wpT by RNAseq, together with their respective controls" showed lower levels of transcripts related to cholesterol synthesis in Dnm2 deleted SCs, and increased expression of genes related to inflammation and cell death. Clodronate liposome injections at the start of phenotype development resulted in delayed recovery, indicating that macrophages may contribute to recovery. Dnm2-null oligodendrocytes, in contrast, had no discernable phenotype, including transferrin uptake (a surprise, I think).

The analysis of the phenotypes of Dnm2 null SCs is well done and well documented. The phenotypes are complex, involving cytokinesis, radial sorting, cell death, and maintenance of myelination, the last of which is quite striking and surprising and worthy of future work. In addition, the demonstration that remyelinating SCs were likely derived from non-recombined SCs was cleverly done and sets the stage for similar studies in other models of demyelination. The authors discuss how loss of DMN2 may affect radial sorting.

The main points that I would like the authors to discuss are:

1) Why DMN2 is needed to maintain myelination (Nrg1/ErbB2 signaling is dispensable), and;

2) How their finding in Dmn2 null SCs relate the neuropathy found in humans with certain dominant DNM2 mutations.

---

## [Author Response]

Reviewer #1 (General assessment and major comments):[…] 1) The studies would be significantly enhanced if the authors were to provide more mechanistic information about why SCs are preferentially dependent on DMN2. Single cell RNAseq of SNs or further interrogation of DMN expression by oligodendrocytes would help in this regard.

Our data show that Schwann cells are dependent on DNM2. We agree that single cell RNAseq of SNs might contribute to a further understanding of this finding and the difference to oligodendrocytes. However, such experiments are a major undertaking and, in our opinion, beyond the scope of our current report. With regard to the expression of dynamins by oligodendrocytes, we have added a statement (including the corresponding references) in the manuscript indicating that oligodendrocytes express also DNM1 and DNM3 (subsection “Dynamin 2 is not essential for oligodendrocyte differentiation and CNS myelination”, first paragraph). To answer whether this fact contributes to our findings concerning these cells will require major, carefully done functional in vivo experiments (e.g. specific inhibition of DNM1/3 in addition to DNM2 deficiency) which are, in our opinion, beyond the scope of our current report.

2) The authors conclude that apoptotic loss of SCs after deletion of DMN2 results in recovery due to de-differentiation of non-recombined SCs. This is supported by the permanent disappearance of YFP-labeled SCs in the nerve. It would be nice if the authors provided additional evidence that the SCs that appear after partial depletion were produced through de-differentiation of mature SCs (rather than possible mobilization of progenitors) using in vivo fate tracing.

Given the genetics of the experimental setting and the currently available tools, it would be technically very challenging to obtain conclusive information with such an in vivo fate tracing. Thus, we have now added a statement related to this aspect in the seventh paragraph of the Discussion section of the manuscript to include the possibility of potential mobilization of progenitors.

3) In Figure 2B what is the purpose of the shaded color in the graph? A scale should be included.

We have removed the shaded color in the graphs in Figure 2B and Figure 3—figure supplement 2 making a scale unnecessary.

4) Figure 3—figure supplement 4B should be included in the main figure to accompany panel Figure 3F.

We would prefer to leave these figures in the arrangement as they are since we feel that the data are better grouped together like this (Figure 3—figure supplement 4B shows the quantification of Figure 3—figure supplement 4A). In our experience, in the special layout of *eLife*, the connection will be obvious for the reader and easy to be followed. However, if the reviewer feels strongly about this point, we will change this.

Reviewer #2:[…] Another series of experiments using tamoxifen induced cut out of dynamin2 in adult nerves showed an intriguing result. This cut out induced a transient neuropathy that was maximal at 14 days after the last injection, followed remarkably by slow remyelination and recovery. The authors provide evidence that Schwann cells with dynamin2 cut out die and are replaced by dynamin2 positive Schwann cells present in the nerves. Two papers suggest that use of this P0ERT-Dnm construct cuts out specifically in myelin Schwann cells in adult mice (Ribeiro et al., 2013; Gomez-Sanchez et al., 2017) adding strength to the authors' suggestion that the remyelination is caused by migration of Remak Schwann cells that fail to die to the demyelinated nerves. There is also evidence for this in older papers from John Griffin's lab, e.g. Murinson, Archer and Griffin, 2005.In summary an excellent paper, fully worthy of publication in eLife.

We have added the references suggested by the reviewer to our manuscript (subsection “Dynamin 2 in Schwann cells is necessary for PNS myelin maintenance”, first paragraph).

Reviewer #3:[…] The main points that I would like the authors to discuss are:1) Why DMN2 is needed to maintain myelination (Nrg1/ErbB2 signaling is dispensable), and;2) How their finding in Dmn2 null SCs relate the neuropathy found in humans with certain dominant DNM2 mutations.

We have expanded the fifth paragraph of the Discussion accordingly, avoiding being too speculative.